# Towards Robust Graph Unlearning via Gradient Consistency Control

## Abstract

Recent graph unlearning models, which aim to efficiently remove undesired data by optimizing a unified objective of forget and retain losses, exhibit a critical vulnerability: their efficacy is severely compromised by inference-time noise attacks on node features. We are the first to diagnose that this fragility stems from a fundamental **gradient inconsistency** problem. Specifically, we theoretically and empirically demonstrate that within the unified optimization objective of graph unlearning, conventional robustness techniques such as adversarial smoothing are counterproductive: they exacerbate the **directional conflict** between the forget and retain gradients, leading to negative interference and failed optimization. To address this, we propose RUNNER, a novel framework for **R**obust graph **UN**learning via gradie**N**t consist**E**ncy Cont**R**ol. RUNNER resolves this conflict through a principled decoupling strategy comprising two core innovations: (1) a decoupled regularization scheme that independently stabilizes gradients from both the forget and retain losses against perturbations, and (2) a gradient alignment objective that penalizes inconsistent gradient between the two losses. Extensive experiments conducted on four real-world datasets demonstrate that RUNNER significantly enhances robustness against noise attacks while maintaining the model's performance under noise-free conditions. Codes are available at https://anonymous.4open.science/r/RUNNER-2FD7.

## 1 Introduction

Graph-structured data are fundamental to represent relational patterns between entities in various real-world applications, such as drug discovery (Rozemberczki et al., 2022; Yu & Gao, 2022; Sypetkowski et al., 2024) and recommendation systems (Shi et al., 2023; Yuan et al., 2025b). To better capture the topological pattern, Graph Neural Networks (GNNs) (Kipf & Welling, 2017; Hamilton et al., 2017) have recently emerged as a crucial tool. With the growing emphasis on privacy protection and introduction of regulatory policies such as GDPR (Voigt & dem Bussche, 2017) and CCPA (CCPA, 2018), there is an increasing urgency to remove privacy-related information from pre-trained graph models. This has motivated a line of research on graph unlearning, aiming to promote *the Right to be Forgotten*. Moreover, graph unlearning is highly valuable for removing inaccurate or outdated information contained in training data (Pawelczyk et al., 2025).

Graph unlearning (Chen et al., 2022; Chien et al., 2023; Wang et al., 2023; Wu et al., 2023; Tan et al., 2024) aims to remove information related to the forget set (edges, nodes, or node features) from a pre-trained graph model while preserving the model utility on retain set. Recently, several approximate-based graph unlearning models (Varshita et al., 2025; Li et al., 2024; Zhang et al., 2025) propose to fine-tune the trained model by incorporating forget loss and retain loss into a unified optimization objective to effectively remove undesired data. Specifically, GNNDelete (Cheng et al., 2023) optimizes Deleted Edge Consistency loss to make the representations of deleted edges resemble those of randomly non-existent edges, and Neighborhood Influence loss to preserve the local subgraph embeddings around the deleted edges, ensuring effective unlearning while maintaining model utility. However, these graph unlearning models (Cheng et al., 2023; Li et al., 2024; Zhang et al., 2025) lack robustness against noise in node features (Madry et al., 2018), such that even a small perturbation applied to the target node during inference can render the unlearning process ineffective. On the other hand, existing adversarial training (Xu et al., 2019; Jin et al., 2020) in graph domains mainly achieves robustness by injecting noise into node features or graph structures at train-

ing stage, e.g., randomly dropping edges (Rong et al., 2020) during training to simulate potential attacks. Such approaches typically rely on prior knowledge to generate adversarial samples. Moreover, achieving robustness in a conventional single-objective training setting (e.g., classification loss) fundamentally differs from the unlearning scenarios, which involves two inherently conflicting objectives: preserving model utility for retain data and ensuring effective forgetting of specific data. We define the unified training objective in the graph unlearning scenario as the two-loss paradigm.

A straightforward approach to bolstering the robustness of graph unlearning is the direct application of smoothing techniques, such as adversarial weight perturbation (Foret et al., 2021). In standard single-objective settings, these methods effectively improve robustness against input noise by smoothing the loss landscape (Du et al., 2022; Kaddour et al., 2022). However, our central finding is that this seemingly straightforward approach is counterproductive in the two-loss paradigm of graph unlearning. As we demonstrate theoretically and empirically (Section 3.3), applying smoothing techniques directly to the unified forget-and-retain objective exacerbates the inherent **directional conflict** between the two constituent gradients. This phenomenon, which we term **gradient inconsistency**, leads to negative interference that not only limits the effectiveness of smoothing but also degrades the model's performance on clean, noise-free data. This insight reveals the core challenge: achieving robustness in the two-loss graph unlearning paradigm is non-trivial and requires a dedicated mechanism to manage gradient conflicts.

To address this fundamental challenge, we propose RUNNER, a novel robust graph unlearning framework designed specifically to manage gradient consistency. Instead of applying smoothing naively, RUNNER introduces a principled decoupling strategy to resolve the underlying gradient conflict. This strategy is realized through two core innovations: (1) a decoupled regularization scheme that mitigates negative interference by independently stabilizing the gradients from both the forget and retain losses against perturbations; and (2) a gradient alignment objective, which further mitigates inconsistency by explicitly penalizing gradient components that become directionally opposed between the two objectives during optimization.

Extensive experiments conducted on four real-world datasets demonstrate that RUNNER significantly enhances robustness against noise attacks while maintaining the model's forget quality under noise-free conditions. Specifically, compared to the baseline that only uses smoothing, RUNNER achieves improvements of 11.95% and 21.94% in AUC and AP on the PubMed forget set under noise attack, respectively. Our contributions are as follows:

- To the best of our knowledge, this is the first attempt to enhance the robustness of graph unlearning to noise attack during inference.
- We are the first to identify that simply applying smoothing techniques to graph unlearning leads to the issue of gradient inconsistency for the goal of robustness.
- We propose RUNNER, which effectively resolves the issue of gradient inconsistency and significantly enhances the robustness of the unlearned model on four real-world datasets.

## 2 PRELIMINARY

**Graph Unlearning.** To achieve efficient graph unlearning while preserving model utility, the unlearning problem is formulated as an optimization task to fine-tune parameters (Edward J et al., 2022; Li et al., 2025; Dai et al., 2024; Wu et al., 2025) from their pretrained model. The optimization objective can be expressed as:

$$\min_\theta \underbrace{\ell_f(\theta \mid D_f)}_{\text{Forget}} + \underbrace{\ell_r(\theta \mid D_r)}_{\text{Retain}}, \tag{1}$$

where $D_f$ and $D_r$ represent the edges to be forgotten and the remaining edges, respectively. $\theta$ represents the learnable parameters, and $\ell_f$ and $\ell_r$ represent the loss functions on $D_f$ and $D_r$, respectively. In this work, we select GNNDelete, Cognac and INPO (Chen et al., 2025), as representatives of the two-loss paradigm. Please see Appendix B for details.

**Lipschitz Constant.** A function $f : \mathbb{R}^n \to \mathbb{R}^m$ is said to be Lipschitz continuous on an input set $\mathcal{X} \subseteq \mathbb{R}^n$ if there exists a bound $L \geq 0$ such that for all $\delta, x \in \mathcal{X}$, and $\delta$ is a small perturbation of $x$, $f$ satisfies:

$$\| f(x + \delta) - f(x) \| \leq L \| (x + \delta) - x \|, \tag{2}$$

where $\| \cdot \|$ represents the norm of a vector and $L$ is a Lipschitz constant of the function, denoted as $Lip(f)$. The Lipschitz constant (Virmaux & Scaman, 2018; Zhao et al., 2021; Juvina et al., 2024; Neacşu et al., 2024; Jia et al., 2024) measures the maximum change in a function's output in response to a small perturbation in its input.

**Task Description.** This work focuses on the robustness of the inference process in the graph unlearning scenario, which specifically aims to enhance the target samples' resilience to noise attacks in practice. This differs from general adversarial defenses (Jin et al., 2020) on graph in two key aspects: 1) unlearning involves two inherently conflicting training losses, and 2) it is trained solely on clean data, without requiring prior knowledge to construct adversarial training samples.

## 3 UNDERSTANDING THE ROBUSTNESS OF GRAPH UNLEARNING TO NOISE

### 3.1 LIMITED NOISE ROBUSTNESS

**(L1) Graph unlearning is sensitive to weight perturbations, making it insufficiently robust to noise.** According to Equation 1, the optimal model on the clean forget set is $\theta_u = \arg\min_\theta \ \ell_f(\theta \mid D_f)$, whereas the optimal model on the noisy data is $\theta'_u = \arg\min_\theta \ \ell_f(\theta \mid D'_f) = \theta_u + \delta$. Therefore, using noisy data for prediction essentially perturbs the weights of the unlearned model. The potential issue is that **the oversensitivity of the weight to perturbation** may lead the unlearned model to be insufficiently robust to noisy input, even under small Gaussian noise perturbations. In such cases, an effective unlearning model should unlearn the forgetting data, while focusing more on robust weight perturbations that be implemented by constructing **worst-case weight perturbations** on unlearning process. This differs from traditional adversarial defense on graphs, which relies on prior knowledge to construct adversarial examples for training.

To demonstrate this issue, we conducted a pilot study from two perspectives: the type of noise and the perturbation of model weights. Figures 1a and 1b show that the forgetting performance on the forget set decreases significantly under different noise types and noise levels, although the extent of the decline is different. The reason for the relatively smaller decrease under salt-and-pepper noise is that using the same setting results in fewer nodes being replaced. Figure 1c indicates that the loss of the unlearned model is **oversensitive** to small perturbations on the weights.

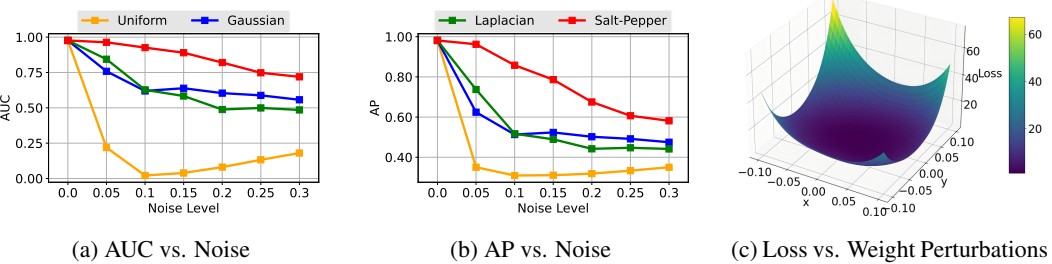

|        (a) AUC vs. Noise        |        (b) AP vs. Noise        |   (c) Loss vs. Weight Perturbations   |

Figure 1: Experimental evidence of insufficiently robust to noise for GNNDelete on PubMed with 0.5% delete ratio. (a)(b) GNNDelete's AUC and AP **on the forget set** at different noise type and level. (c) The prediction loss landscape of GNNDelete, and the 3D loss landscape is defined as $z = \ell(\theta + x \cdot r_1 + y \cdot r_2)$, with $\theta$ representing the unlearned model. $r_1$ and $r_2$ are vectors representing the perturbation directions, and $x, y$ denote the magnitudes of the perturbations.

### 3.2 SMOOTHING METHODS TO IMPROVE UNLEARNING ROBUSTNESS

To enhance robustness of graph unlearning model, adopting adversarial optimization (Madry et al., 2018; Foret et al., 2021) is a straightforward approach, $\delta$ **denotes the perturbation applied to the parameters**, and the optimization objective can be expressed as following:

$$\min_\theta \ \max_{\|\delta\|_p \leq \rho} \ \ell_f(\theta + \delta \mid D_f) + \ell_r(\theta + \delta \mid D_r), \tag{3}$$

where the hyper-parameter $\rho$ represents the perturbation radius, which controls limit the ability to disrupt the unlearned model. $\| \cdot \|_p$ denotes the norm $\ell_p$, with $p = 2$ as the default setting.

Adversarial optimization is actually achieved through gradient or curvature smoothing to enhance robustness against noise attack during inference. The theoretical analysis can be found in Section 4.1. We introduce four smoothing methods to improve unlearning robustness here:

(1) **Weight Averaging (WA)** (Izmailov et al., 2018), which enforces smoothness by dynamically updating weights during the training process. The weight update is $\bar{\theta}_{t+1} = (1 - k) \cdot \bar{\theta}_t + k \cdot \theta_{t+1}$, where $k = 1/(t + 1)$. Here, $t$ denotes the $t$-th training epoch.

(2) **Randomized Smoothing (RS)** (Cohen et al., 2019; Ji et al., 2024), which adds noise sampled from the Gaussian distribution $\mathcal{N}(0, \sigma^2)$ to the weights during the training process. The loss with RS is expressed by:

$$\ell^{\text{RS}}(\theta) = \ell_f^{\text{RS}}(\theta) + \ell_r^{\text{RS}}(\theta) = \mathbb{E}_{\delta \sim \mathcal{N}(0,\sigma^2)}\left[\ell_f(\theta + \delta)\right] + \mathbb{E}_{\delta \sim \mathcal{N}(0,\sigma^2)}\left[\ell_r(\theta + \delta)\right]. \tag{4}$$

(3) **Gradient Penalty (GP)** (Tong et al., 2020; Liu et al., 2024; Zhao et al., 2024), where gradients are used as a regularization term to improve robustness, as shown in Equation 8. The optimization objective is given by:

$$\ell^{\text{GP}}(\theta) = \ell_f(\theta) + \ell_r(\theta) + \rho\|\nabla_\theta(\ell_f(\theta) + \ell_r(\theta))\|_2. \tag{5}$$

(4) **Regularization of the Curvature (CR)** (Dauphin et al., 2024), which achieves robustness to weight perturbation by explicitly penalizing the curvature of the loss. The optimization objective can be expressed by:

$$\ell^{\text{CR}}(\theta) = \ell_f(\theta) + \ell_r(\theta) + \gamma(\|\nabla_\theta(\ell_f(\theta + \delta) + \ell_r(\theta + \delta)) - \nabla_\theta(\ell_f(\theta) + \ell_r(\theta))\|_2). \tag{6}$$

$\sigma, \rho, \gamma > 0$ are hyper-parameters. These smoothing methods essentially aim to make the gradient approach zero or to make the curvature approach zero, thereby achieving stability of the loss landscape w.r.t. $\theta$.

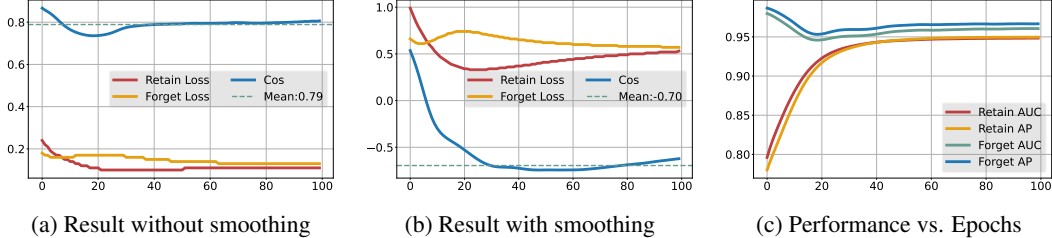

(a) Result without smoothing      (b) Result with smoothing      (c) Performance vs. Epochs

Figure 2: Experimental evidence of inconsistent gradients and difficult unlearning for GNNDelete on PubMed. (a)(b) The relationship between loss and gradient cosine similarity (Cos) across epochs without CR and with CR. (c) The performance across different epochs for GNNDelete.

### 3.3 GRADIENT INCONSISTENCY

**(L2) Smoothing methods cause inconsistent gradients, making smoothing and unlearning ineffective.** Smoothing methods in graph unlearning two-loss paradigm 1 would introduce the issue of inconsistent gradients, making smoothing and unlearning difficult to optimize. Gradient Inconsistency refers to the situation where gradients $(g_r, g_f)$ from the utility-preserving loss and the forgetting loss are **directionally opposed** (i.e., $cos(g_r, g_f) = \frac{<g_f, g_r>}{||g_f|| \cdot ||g_r||} < 0$) after smoothing. The theoretical explanation for the gradient inconsistency caused by smoothing methods is provided in Theorem 3 and Appendix E, which undermines the effectiveness of smoothing techniques.

An ideal unlearning optimization process (i.e., $\theta_{t+1} = \theta_t - \eta_1 g_f^t - \eta_2 g_r^t$) would have gradients in the same direction for utility-preserving loss and forgetting loss. Here, $t$ denotes the $t$-th training epoch, and $\eta_1$ and $\eta_2$ are hyper-parameters. For simplicity, assuming the model is locally linear (Ross & Doshi-Velez, 2018; Zhang et al., 2019; Wang et al., 2025), the loss change on the retain data can be expressed as $\ell_r(\theta + \eta_1 g_f) - \ell_r(\theta) \approx \eta_1\langle g_f, g_r \rangle$. A more rigorous proof is given in Theorem 3. We can conclude that a negative $\langle g_f, g_r \rangle$ implies that the reverse gradient would have **adverse effect** on the loss of the retain set. Obviously, Gradient inconsistency also makes unlearning ineffective.

We support this finding by illustrating the relationship between loss and the direction consistency (computed by $cos(g_f, g_r)$) of the two gradients. As shown in Figures 2a and 2b, during the first 20 epochs, the unlearned model with smoothing techniques exhibits a significant increase in gradient inconsistency, with the average cosine similarity (Mean) of gradients over the subsequent 80 epochs decreasing from 0.79 to $-0.70$. The retain loss of the method with the smoothing regularization increases at a significantly **higher rate** than that of the method without regularization, indicating that smoothing regularization amplifies the negative interference between the two conflicting losses. Figure 2c indicates that the misalignment in the gradient direction makes it difficult to maintain performance on the forget set under noise-free. Therefore, while adopting smoothing methods to enhance robustness, it is important to solve gradient inconsistency.

# 4    ENHANCING UNLEARNING ROBUSTNESS BY GRADIENT CONSISTENCY

## 4.1    WHY DOES ADVERSARIAL OPTIMIZATION IMPROVE ROBUSTNESS TO NOISE?

Here, we theoretically demonstrate the effectiveness of smooth methods to input noise.

**Adversarial optimization is essentially gradient or curvature smoothing.** The essence of the optimization object Equation 3 is to learn a model robust to weight perturbations to counteract noise. We start the analysis with $\ell_f(\theta + \delta \mid D_f)$ as the basis. The inner maximization can be solved in closed form using linear approximation:

$$\delta^*(\theta) := \underset{\|\delta\|_p \leq \rho}{\arg\max} \ \ell_f(\theta + \delta) \approx \underset{\|\delta\|_p \leq \rho}{\arg\max} \ \ell_f(\theta) + \delta^\top \nabla_\theta \ell_f(\theta) = \rho \frac{\nabla_\theta \ell_f(\theta)}{\|\nabla_\theta \ell_f(\theta)\|_2}. \tag{7}$$

The outer minimization now is:

$$\min_\theta \ell_f(\theta + \delta \mid D_f) = \min_\theta \ell_f(\theta + \rho \frac{\nabla_\theta \ell_f(\theta)}{\|\nabla_\theta \ell_f(\theta)\|_2}) \approx \min_\theta \ell_f(\theta) + \rho \frac{\nabla_\theta \ell_f(\theta)^\top \nabla_\theta \ell_f(\theta)}{\|\nabla_\theta \ell_f(\theta)\|_2}$$
$$= \min_\theta \ell_f(\theta) + \rho \|\nabla_\theta \ell_f(\theta)\|_2. \tag{8}$$

Therefore, the optimization of unlearned model's robustness to weight perturbations essentially approximately equal to reduce the gradient and curvature of loss, i.e., smoothing the gradient and curvature of the forget loss landscape w.r.t. $\theta$.

**Lemma 1** (Lipschitz Bound (Juvina et al., 2024)). *Assuming the activation function (represents in $\rho(\cdot)$) is ReLU with a Lipschitz bound of $Lip(\rho) = 1$, then the cumulative Lipschitz bound of the entire GNN, Lip(f), satisfies:*

$$Lip(f) \leq \prod_{i=1}^{L} ||W_i||_2, \tag{9}$$

*where $W_i$ denotes weights of i-th layer in GNN, and $|| \cdot ||_2$ represents the spectral norm.*

**Gradient or curvature smoothing is equivalent to reducing the Lipschitz bound.** Based on Equation 2 and Equation 9, we conclude that unbounded and unstable gradient or weight can make the model's output highly sensitive to small input perturbations, i.e., graph unlearning model is not robust to input noise. Therefore, smoothing the gradient can yield a bounded and stable Lipschitz bound, achieving robustness to input noise.

In summary, adversarial optimization improves the model's robustness to noise by smoothing the gradient and curvature of each loss. However, two-loss paradigm of graph unlearning can lead to gradient inconsistency issues, which limit the smoothing capability.

## 4.2    GRADIENT CONSISTENCY CONTROL AS OPTIMIZATION

As illustrated in Section 3.3, inconsistent gradients ($cos(\nabla_\theta \ell_f(\theta), \nabla_\theta \ell_r(\theta)) < 0$) have adverse effect on smoothing and performance. To make graph unlearning both robust to noise and capable of maintaining the forget quality under noise-free, we propose the following two **simple yet effective** methods based on the CR:

(1) Gradients Separation Control (GSC), which mitigates negative gradient interference by independently regulating the perturbation stability associated with $\ell_f$ and $\ell_r$, and this is expressed by:

$$\ell^{\text{GSC}}(\theta) = \ell_f(\theta) + \ell_r(\theta) + \gamma_1(\|\nabla_\theta\ell_f(\theta+\delta) - \nabla_\theta\ell_f(\theta)\|_2) + \gamma_2(\|\nabla_\theta\ell_r(\theta+\delta) - \nabla_\theta\ell_r(\theta)\|_2), \quad (10)$$

where $\gamma_1, \gamma_2$ are used to control the robustness of the forget loss and the retain loss against gradient perturbations, respectively. According to Theorem 3 and 5 in Appendix E, GSC avoids gradient direction flip (i.e., inconsistent gradients) by controlling the lower bound of a key term during smoothing process. Figure 5 illustrates the comparison of the key term.

(2) Human-aware Alignment (HA), which assigns different smoothness tolerances to data with varying levels of consistency on gradient direction using a loss similar to KTO (Ethayarajh et al., 2024; Haldar et al., 2025), and HA is given by:

$$\ell^{\text{HA}}(\theta) = \ell_f(\theta) + \ell_r(\theta) + v(\ell_f; \gamma_1, \gamma_2, \alpha) + v(\ell_r; \gamma_1, \gamma_2, \alpha), \quad (11)$$

$$v(z; \gamma_1, \gamma_2, \alpha) = \begin{cases} \gamma_1\|\nabla_\theta z(\theta+\delta) - \nabla_\theta z(\theta)\|_2^\alpha & \text{if aligned,} \\ \gamma_2\|\nabla_\theta z(\theta+\delta) - \nabla_\theta z(\theta)\|_2^\alpha & \text{if unaligned,} \end{cases} \quad (12)$$

where $z$ is $l_f$ or $l_r$ and $\alpha$ is a hyper-parameter. When the loss is convex shape (i.e., $\alpha > 1$), the risk-aversion property of the human-aware loss makes the model more sensitive to weight fluctuations that exceed a certain range. In other words, after weight perturbation, the model tolerates small fluctuations in the loss landscape. Please see Appendix E.4 for the theoretical proof and details.

Aligned data are characterized by **the consistency in gradient direction** induced by perturbations across both utility-preserving loss and forgetting loss, i.e., $cos(\nabla_\theta\ell_f(\theta+\delta), \nabla_\theta\ell_r(\theta+\delta)) > 0$. $\gamma_1, \gamma_2$ control the tolerance towards aligned and unaligned data, respectively. $\gamma_2 > \gamma_1$ indicates that consistent gradients are allowed to take effect, while inconsistent gradients are suppressed as much as possible. This further alleviates the negative impact of inconsistent gradients for smoothing.

## 5 EXPERIMENTS

### 5.1 EXPERIMENT SETUPS

**Datasets.** To thoroughly validate the effectiveness of our model and ensure a comprehensive generalization evaluation, we used four real-world datasets (Bojchevski & Günnemann, 2018; Hu et al., 2020): Cora, PubMed, DBLP, and CS. The table in Appendix C summarizes the statistical details of the graph datasets.

**Baseline Models.** In our experiments, we select 5 advanced graph unlearning models and 4 smoothing optimization techniques as illustrated in Section 3.2 as baselines for comparison. These baselines include Retrain, GIF, UtU, GNNDelete and INPO.

**Evaluation Metrics.** To measure the effectiveness of our model, we use model utility and forget quality as metrics. The model utility refers to its ability to maintain the original inference capability after unlearning, measured by AUC and AP on the retain set. The forget quality measured by AUC and AP on the forget set. $AP = \sum_n (R_n - R_{n-1}) \cdot P_n$ where $P_n$ and $R_n$ are the precision and recall at the $n$-th threshold. AUC is the area under the Receiver Operating Characteristic Curve. To evaluate the model's robustness against noise attacks, we primarily compare its effectiveness under various types and levels of noise perturbations, include AUC/AP on the forget and retain set.

**Setups.** We evaluate the effectiveness of our model on edge unlearning tasks. We perform experiments on following setting: (1) The forget set $D_f$ is constructed by randomly sampling edges from outside the 2-hop enclosing subgraph of test set and (2) Performing noise attacks on the node features of the endpoints of each edge to be deleted. To perform edge unlearning tasks, the proportion of edges we delete is $0.5 - 5.0\%$. We conducted all experiments 5 times and reported average value.

### 5.2 OVERALL PERFORMANCE

**Analysis on the baselines.** In Figure 3a, we observe that current graph unlearning models (e.g., GNNDelete, INPO) are highly non-robust to noise perturbations. Moreover, simply incorporating

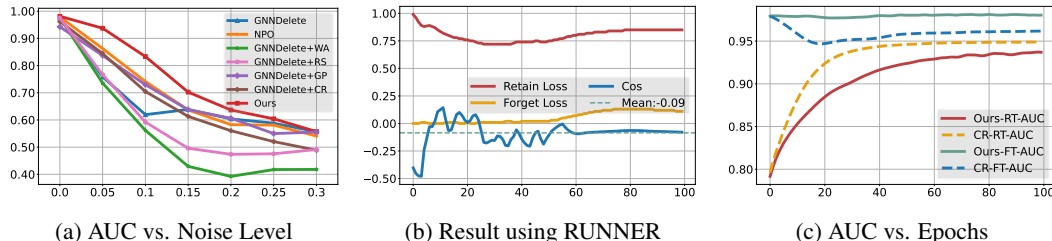

(a) AUC vs. Noise Level      (b) Result using RUNNER      (c) AUC vs. Epochs

Figure 3: Graph Unlearning robustness comparison for different methods on PubMed dataset. (a) AUC on the forget set vs. different noise level. (b) The relationship between loss and gradient similarity across epochs for RUNNER. (c) The AUC across different epochs for RUNNER.

Table 1: Unlearning robustness comparison of RUNNER and different smoothness optimization-based models on PubMed and Cora under noise attack (Type: Gaussian, Level: 0.05) settings. In each column, the best result is indicated in red, while the runner-up result is marked with blue. Task: Link Prediction. RUNNER-G and RUNNER-N refer to the models using Loss $\mathcal{L}_f^{\mathrm{G}}$ and $\mathcal{L}_f^{\mathrm{N}}$, respectively.

| Model | PubMed | | | | Cora | | | | Avg |
|---|---|---|---|---|---|---|---|---|---|
| | $D_r$ | | $D_f$ | | $D_r$ | | $D_f$ | | |
| | AUC | AP | AUC | AP | AUC | AP | AUC | AP | |
| Retrain | 0.9538 | 0.9595 | 0.4631 | 0.4325 | 0.8997 | 0.8854 | 0.3184 | 0.3792 | 0.6627 |
| GIF | 0.9461 | 0.9489 | 0.5406 | 0.4723 | 0.8690 | 0.8505 | 0.2991 | 0.3837 | 0.6638 |
| UtU | 0.9458 | 0.9490 | 0.5359 | 0.4690 | 0.8689 | 0.8506 | 0.3092 | 0.3885 | 0.6646 |
| INPO | 0.9459 | 0.9469 | 0.8626 | 0.7632 | 0.9016 | 0.8867 | 0.5754 | 0.4901 | 0.7966 |
| GNNDelete | 0.9464 | 0.9460 | 0.7584 | 0.6244 | 0.8986 | 0.8745 | 0.5141 | 0.4568 | 0.7524 |
| GNNDelete+WA | 0.9436 | 0.9419 | 0.7357 | 0.6166 | 0.8786 | 0.8460 | 0.4617 | 0.4322 | 0.7320 |
| GNNDelete+RS | 0.9469 | 0.9448 | 0.7665 | 0.6476 | 0.8820 | 0.8474 | 0.3739 | 0.3977 | 0.7259 |
| GNNDelete+GP | 0.9108 | 0.9121 | 0.8371 | 0.7550 | 0.7843 | 0.7693 | 0.5678 | 0.4856 | 0.7528 |
| GNNDelete+CR | 0.9469 | 0.9466 | 0.8443 | 0.7643 | 0.9132 | 0.8869 | 0.5336 | 0.4670 | 0.7879 |
| **RUNNER-G** | 0.9378 | 0.9381 | 0.9452 | 0.9320 | 0.9021 | 0.8836 | 0.6132 | 0.5145 | 0.8333 |
| **RUNNER-N** | 0.8811 | 0.8715 | 0.9497 | 0.9559 | 0.9025 | 0.8870 | 0.6940 | 0.5725 | 0.8393 |

smoothing methods has limited effectiveness and some smoothing methods (WA, RA) are even ineffective in two-losses graph unlearning paradigm. Our model consistently outperforms the baselines across all perturbation levels in such paradigm.

**The effectiveness of RUNNER.** In Table 1, we summarize the overall performance of RUNNER and the baselines. We find that our method significantly enhances the robustness of graph unlearning to noise compared to existing baselines and self-designed smoothing methods. For instance, compared to GNNDelete+CR, RUNNER-G achieves improvements of 11.95% and 21.94% in AUC and AP on the forget set of PubMed under noise attack, respectively. Similarly, on the Cora dataset, the improvements are 14.92% and 10.17%. Also, the AUC and AP on the retain set are maintained(except for RUNNER-N on PubMed), and the average of all metrics also shows significant improvement(5.76%, 6.52%). Overall, our models achieves state-of-the-art performance on forget quality metrics for noise attack while maintaining the model's performance under noise-free conditions (detailed analysis are deferred to Section 5.3.), which demonstrates the effectiveness of gradient consistency control. The results of RUNNER-G and RUNNER-R demonstrate that our method can be easily extended to existing graph unlearning frameworks.

**The reasons for the effectiveness of RUNNER.** As shown in Figure 2b and 2c, the single use of gradient smoothing can lead to inconsistent gradient directions (larger reverse similarity), negatively impacting the optimization process. RUNNER **avoids the negative impact of inconsistency on optimization** by encouraging perturbation-based unlearning using data samples with consistent gradient directions. Figure 3b show that, compared to GNNDelete+CR, RUNNER significantly re-

Table 2: Unlearning robustness of RUNNER-G on PubMed under different noise attack settings (Level: 0.05). Task: Link Prediction. Robustness in pink is significantly better than ones in gray.

| Model | GNNDelete | | | | RUNNER | | | |
|---|---|---|---|---|---|---|---|---|
| | $D_r$ | | $D_f$ | | $D_r$ | | $D_f$ | |
| | AUC | AP | AUC | AP | AUC | AP | AUC | AP |
| Uniform Noise | 0.9174 | 0.8974 | 0.0685 | 0.3185 | 0.9368 | 0.9318 | 0.5047 | 0.5270 |
| Laplacian Noise | 0.9436 | 0.9420 | 0.7374 | 0.6183 | 0.9373 | 0.9383 | 0.9403 | 0.9238 |
| Salt-and-Pepper Noise | 0.9580 | 0.9551 | 0.9325 | 0.8803 | 0.9451 | 0.9434 | 0.9751 | 0.9736 |
| Average | 0.9397 | 0.9315 | 0.5795 | 0.6057 | 0.9397 | 0.9378 | 0.8067 | 0.8081 |

Table 3: Ablation results of RUNNER on PubMed under noise attack (Type: Gaussian, Level: 0.05, 0.10) settings. In each column, the best result is indicated in red, while the runner-up result is marked with blue. Task: Link Prediction. Avg denotes the average performance on the forget set.

| Model | Level=0.05 | | | | | Level=0.10 | | | | |
|---|---|---|---|---|---|---|---|---|---|---|
| | $D_r$ | | $D_f$ | | | $D_r$ | | $D_f$ | | |
| | AUC | AP | AUC | AP | Avg | AUC | AP | AUC | AP | Avg |
| GNNDelete | 0.9464 | 0.9460 | 0.7584 | 0.6244 | 0.6914 | 0.9220 | 0.9205 | 0.6195 | 0.5131 | 0.5663 |
| RUNNER-CR | 0.9469 | 0.9466 | 0.8443 | 0.7643 | 0.8043 | 0.9253 | 0.9265 | 0.7045 | 0.5761 | 0.6403 |
| RUNNER-GSC | 0.9466 | 0.9466 | 0.9293 | 0.9013 | 0.9153 | 0.9287 | 0.9309 | 0.8196 | 0.7012 | 0.7604 |
| RUNNER-HA | 0.9378 | 0.9381 | 0.9452 | 0.9320 | 0.9386 | 0.9269 | 0.9308 | 0.8604 | 0.7690 | 0.8147 |

duces the larger reverse similarity in the last 80 epochs (from $-0.70$ to $-0.09$), which explains why our model exhibits robustness. The AUC curve, which does not show a decline, further indicates a reduced impact on forget quality under noise-free. This is evidenced by the results in Figure 3c.

**Robustness against various types of noise.** As shown in Table 2, RUNNER significantly enhances robustness against noise attacks under various noise attacks. Specifically, compared to GNNDelete, RUNNER improves AUC and AP in the forget set by 22.72% and 20.24%, respectively. It is worth noting that although the robustness is significantly enhanced under Uniform Noise attacks, the ability to resist such noise perturbations remains limited. Compared to Laplacian or Salt-and-Pepper noise, which apply more concentrated or sparse perturbations, Uniform Noise tends to evenly disturb the entire feature space. This causes more severe and unpredictable degradation in model performance, especially for tasks like link prediction.

## 5.3 ABLATION STUDY

Here we empirically dissect the contribution of (1) Smoothing method, (2) Gradients separation control, and (3) Human-aware alignment in our model. We proposed three ablation models, respectively:

(1) **RUNNER-CR**, which introduces curvature regularization into the initial graph unlearning model (e.g., GNNDelete) to enhance robustness.

(2) **RUNNER-GSC**, which introduces GSC on RUNNER-CR and mitigates negative gradient interference by independently regulating the perturbation-based stability associated with $\ell_f$ and $\ell_r$.

(3) **RUNNER-HA**, which introduces HA on RUNNER-GSC, assigns different smoothness tolerances to data with varying levels of consistency in the gradient direction.

**Ablation results.** In Table 3, we report the ablation results. By comparing RUNNER-CR and GNNDelete, we discover that it can significantly enhance the robustness to noise attack, but it is still far from a satisfactory result. Furthermore, the comparison between RUNNER-CR and RUNNER-GSC implies that GSC promotes a separate fine-grained control by independently regulating the stability based on perturbations in $\ell_f, \ell_r$ and mitigates negative gradient interference. RUNNER-GSC im-

proves the average forgetting performance by 13.80% at Level=0.05 and by 18.76% at Level=0.10, respectively. RUNNER-HA consistently outperforms baselines, and also outperforms RUNNER-GSC, especially under the Level=0.10 setting. Compared to RUNNER-GSC, RUNNER-HA improves the average forgetting performance by 2.55% at Level=0.05 and by **7.14% at Level=0.10**, respectively. In summary, results demonstrate that our methods effectively enhances robustness to noise attack.

**The impact of noise level.** From the above comparison, it can be seen that HA performs more effectively under settings with higher noise levels. Under large perturbations, samples with inconsistent gradient changes become more prevalent. HA effectively suppresses the model's gradient drift in incorrect directions by imposing stronger penalties on these inconsistent gradients. Therefore, by employing larger values of $\gamma_2(\gamma_2 : \gamma_1 = 3 : 1)$ and $\alpha \geq 1$, actively shrinking the inconsistent components in the gradient space helps HA achieve better robustness against large noise. HA avoids the potential deterioration of unlearning effectiveness caused by the simplistic use of gradient rectification or clipping methods.

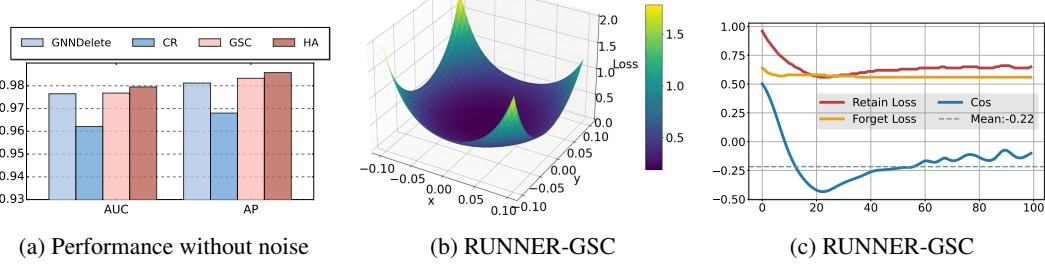

(a) Performance without noise      (b) RUNNER-GSC      (c) RUNNER-GSC

Figure 4: Experimental evidence of the effectiveness of the ablation models. (a) Performance comparison between ablation models and the baseline on the forget set under noise-free conditions. (b) The prediction loss landscape of RUNNER-GC. (c) The relationship between loss and gradient similarity across epochs using RUNNER-GSC.

**A comparison of the smoothness between GSC and HA.** Overall, HA further enhances robustness by improving the optimization process. As shown in Figure 4a, both ablation models lead to improved forget quality under no-noise conditions. From the comparison between Figure 4b and 1c, we observe that RUNNER effectively achieves robustness against weight perturbations, with the fluctuation range reduced from [0, 60] to [0, 2], under the same scaling factor and experimental settings. From the comparison between Figure 4c and 3b, HA further reduces gradient inconsistency, with the mean decreasing from $-0.22$ to $-0.09$.

### 5.4 PARAMETER SENSITIVITY STUDY

Here, we mainly study how the choice of hyper-parameter $\alpha$ affects the model's utility and the robustness of unlearning. As shown in the table of Appendix H, $a = 2$ achieves the best trade-off, demonstrating that $a > 1$ enables the model to reject large perturbations while allowing small fluctuations, thereby avoiding overfitting to noise and preserving performance on the retain set. This result is fully consistent with the theory we propose. Moreover, an overly large $\alpha$ may undermine the model's robustness. Throughout all experiments in this work, we consistently set $\alpha = 2$. For the impact of $\gamma_1$ and $\gamma_2$ on the model, please refer to Appendix H.2.

### 6 CONCLUSION

In this work, we are the first to propose the robustness of graph unlearning against inference-time noise attacks. Specifically, we are the first to identify that simply applying smoothing techniques leads to the issue of gradient inconsistency in two-loss graph unlearning paradigm, which is supported by both experimental and theoretical evidence. We propose RUNNER, which mitigates negative interference between gradients by gradient separation control and human-aware alignment loss. Extensive experiments conducted on four real-world datasets demonstrate that RUNNER significantly enhances robustness against noise attacks during inference phase while maintaining the model's performance under noise-free conditions and exhibits strong scalability.

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

## A  RELATED WORK

**Graph Unlearning**. Retraining (Liu et al., 2022) refers to training the model from scratch to forget specific edges rather than fine-tuning parameters, and it is time consuming and resource intensive. GraphEraser (Chen et al., 2022) attempts to achieve graph unlearning by utilizing graph partitioning and efficient retraining, but it supports only node unlearning. GraphEditor (Cong & Mahdavi, 2023) offers a closed-form solution for linear GNNs to ensure information removal, with fine-tuning improving model utility, but it is limited to linear structures and not designed for graph-structured data. GIF (Wu et al., 2023) uses influence functions to estimate parameter changes for edge unlearning but performs poorly on the forget set and cannot achieve true unlearning. Overall, these methods have room for improvement in either performance or efficiency.

Recently, several approximate-based graph unlearning models (Cheng et al., 2023; Li et al., 2024; Tan et al., 2024) propose to fine-tune the trained model by incorporating forget loss and retain loss into a unified optimization objective to effectively remove undesired data. Specifically, GNNDelete (Cheng et al., 2023) achieves unlearning by minimizing two losses: one approximates the representation of edges to be forgotten to those that did not exist in the pretrained model, and the other keeps the neighbors' representations minimally changed. However, these two losses are in conflict and lack robustness against weight perturbations. MEGU (Li et al., 2024) propose a new mutual evolution paradigm that simultaneously evolves the model utility and forget capacities of graph unlearning, which is achieved through two losses. Compared to GNNDelete (Cheng et al., 2023), UtU (Tan et al., 2024) only uses the graph after edge deletion for a single inference. However, these models are vulnerable to noise attacks due to the joint loss failing to smooth weight perturbations. In this work, we aim to enhance the robustness of the unlearned model to the noise attack.

**Large Language Model Unlearning**. Gradient Ascent (Yuan et al., 2025a) employees fine-tuning to maximize cross-entropy loss on the forget set to unlearn information. INPO (Chen et al., 2025; Zhang et al., 2024a) adjusts offline DPO (Rafailov et al., 2023) to reduce the likelihood of the forget set, avoiding the complexity of learning a reward function like RLHF (Dai et al., 2024). SimNPO (Fan et al., 2024) propose a simple yet effective unlearning optimization framework to remove the reliance on a reference model, which directly uses the edges to be forgotten as negative samples. To verify the scalability of our method, we introduce loss of INPO as our forget loss.

**Attack-resistant adversarial optimization**. Adversarial training to defend against input-level adversarial attacks (Liu et al., 2024; Wei et al., 2023; Zhang et al., 2024b) has achieved strong robustness in the single-loss setting. Relearning defense (Fan et al., 2025) explores enhancing the robustness of unlearning in large language models against relearning attacks through sharpness-aware minimization. However, they solely focus on enhancing robustness of forget quality by minimizing weight perturbations in the single-loss paradigm. In this work, we simultaneously consider the robustness of forget quality and model utility against noise attacks, and identify and resolve gradient

inconsistencies caused by strong coupling relationships between graph nodes in the multi-objective graph unlearning paradigm.

## B  Loss in Two-loss Paradigm

In this work, we select GNNDelete and INPO, which is a preference-based optimization framework, as representatives of the two-loss paradigm. For GNNDelete,

$$\mathcal{L}_f^{\mathrm{G}} = \mathbf{MSE}\left(\{[h_u^{l'}; h_v^{l'}] \mid e_{uv} \in \mathcal{D}_f\}, \{[h_u^l; h_v^l] \mid u, v \in \mathcal{V}\}\right), \tag{13}$$

$$\mathcal{L}_r = \mathbf{MSE}\left(\big\|_w \{h_w^{l'} \mid w \in S_{uv}^l \setminus e_{uv}\}, \big\|_w \{h_w^l \mid w \in S_{uv}^l\}\right), \tag{14}$$

where $h_u^{l'}, h_v^{l'}, h_w^{l'}$ and $h_u^l, h_v^l, h_w^l$ represent the node representations of the unlearned model and the pre-trained model, respectively. **MSE** is Mean Squared Error loss on training sets and $S_{uv}^l$ is the local subgraph of edge $(u, v)$. $[\cdot]$ and $\|$ denote the concatenation of the features of the two nodes connected by an edge and the features of the nodes within the local subgraph, respectively. Additionally, we introduce a effective preference optimization-based graph unlearning loss $\ell_f^N$ (INPO) (Chen et al., 2025; Zhang et al., 2024a; Fan et al., 2024) from the LLM Unlearning (Yuan et al., 2025a; Sungmin et al., 2025) domain for more comprehensive comparison, and it is:

$$\mathcal{L}_f^{\mathrm{N}}(\theta) = \frac{2}{\beta}\mathbb{E}_{(x,y)\sim\mathcal{D}_f}[log(1 + \frac{\pi_\theta(y \mid x)}{\pi_{ref}(y \mid x)})^\beta], \tag{15}$$

where $\pi_\theta(y \mid x)$ and $\pi_{ref}(y \mid x)$ represent the predicted probability for edges on $D_f$ from the unlearned model and the pre-trained model, respectively. Let $x$ denote the learned edge representation and $y$ denote negative preference of this edge, and $\beta$ is a hyperparameter.

## C  Experiment Setups

### C.1  Noise Attacks

Here, we provide a detailed introduction to the design of four types of noise attacks for edge unlearning: (1) Uniform Noise: We add noise from uniform distribution $\epsilon \sim U(0, 1)$ to the node features of the endpoints of each edge to be forgotten, its probability density function (PDF) is given by:

$$p(x) = \begin{cases} 1, & \text{if } 0 \le x \le 1 \\ 0, & \text{otherwise.} \end{cases} \tag{16}$$

(2) Gaussian Noise: We add noise from a Gaussian distribution $\epsilon \sim \mathcal{N}(\mu, \sigma^2)$ to the node features of the endpoints of each edge to be forgotten, its PDF is given by:

$$p(x) = \frac{1}{\sqrt{2\pi\sigma^2}}e^{-\frac{(x-\mu)^2}{2\sigma^2}}, \tag{17}$$

where $\mu$ is the mean and $\sigma^2$ is the variance of the distribution.

(3) Laplace Noise: We add noise from a Laplace distribution $\epsilon \sim \text{Laplace}(\mu, \sigma)$ to the node features of the endpoints of each edge to be forgotten, its PDF is given by:

$$p(x|\mu, \sigma) = \frac{1}{2\sigma}\exp\left(-\frac{|x-\mu|}{\sigma}\right), \tag{18}$$

where $\mu$ is the location parameter and $\sigma > 0$ is the scale parameter of the distribution.

(4) Salt and Pepper Noise: We apply salt and pepper noise to the node features of the endpoints of each edge to be forgotten. This type of noise randomly changes the features of some nodes to the minimum or maximum value. The effect of this noise can be described as follows:

- A fraction $p_s$ of the nodes is set to the maximum value (salt noise).

- A fraction $p_p$ of the nodes is set to the minimum value (pepper noise).

Here, $p_s$ and $p_p$ represent the probabilities of node features being affected by salt and pepper noise, respectively.

To represent the noise level, we set different values of $\sigma, p_s, p_p \in [0.05, 0.10, 0.15, 0.20, 0.25, 0.30]$. Larger values indicate stronger noise perturbations.

## C.2 HYPER-PARAMETERS SETTINGS

For the dataset, we randomly split it into training set, validation set, and test set in the ratio of 0.9:0.05:0.05, following the same setting as GNNDelete (Cheng et al., 2023). Following the architectural design of GNNDelete (Cheng et al., 2023), the RUNNER model employs **two** GCN layers as its backbone. For the fine-tuning parameters, we add new trainable additional parameters after each layer, and use model-agnostic layer-wise deletion operator. In the implementation of the GCN layer, we adopt the most commonly used massage passing neural network (MPNN) framework. We set the hidden layer dimension to 128. The dimension of the additional trainable parameters introduced after each layer is also 128. In the training stage, we set the initial learning rate to 1e-3, weight decay to 0.0005, and use the Adam optimizer. The model is fine-tuned for 100 epochs. The gradient perturbation radius $\rho$ is set to 0.1. It should be noted that all the experiments in the Appendix used a ratio of $\gamma_1 : \gamma_2 = 50 : 150$, and $\alpha = 2$. The ratio of $\gamma_1 : \gamma_2 = 0.2 : 1$ in main text for Cora balances the performance of both RUNNER-G and RUNNER-N.

Table 4: Statistics of evaluated datasets.

|          | Cora    | PubMed | DBLP    | CS      |
|----------|---------|--------|---------|---------|
| # Nodes  | 19,793  | 19,717 | 17,716  | 18,333  |
| # Edges  | 126,842 | 88,648 | 105,734 | 163,788 |

# D ALGORITHM FOR RUNNER

Algorithm 1 outlines the core steps of RUNNER.

# E PROOF AND ELABORATION OF THEOREM 3

## E.1 NOTATION AND ASSUMPTIONS

Let $\ell_f, \ell_r : \mathbb{R}^d \to \mathbb{R}$ be real-valued functions that are at least three times differentiable in a neighborhood of the point $\theta$. Define

$$g_f := \nabla \ell_f(\theta), \qquad g_r := \nabla \ell_r(\theta).$$

We impose the following assumptions on the random perturbation $\delta \in \mathbb{R}^d$:

(A1) $\mathbb{E}[\delta] = 0$ and $\mathrm{Cov}(\delta) = \Sigma$.

(A2) There exists a constant $C_3 > 0$ such that $\mathbb{E}\|\delta\|^3 \leq C_3 \|\Sigma\|^{3/2}$.

(A3) For $\ell \in \{\ell_f, \ell_r\}$ there exists a constant $M_\ell > 0$ such that the third-order derivative tensor is bounded in the neighborhood:

$$\|\nabla^3 \ell(\theta')\|_{\mathrm{op}} \leq M_\ell, \qquad \forall \, \theta' \text{ in the neighborhood,}$$

where $\|\nabla^3 \ell\|_{\mathrm{op}}$ denotes the operator norm of the third-order tensor.

In the following, $\|\cdot\|$ denotes the $\ell_2$ norm for vectors.

---

**Algorithm 1** Consistency-enhanced Graph Unlearning

---

1: **Input:** Pre-trained model $\theta$, forget loss $\ell_f$, retain loss $\ell_r$, unlearning epochs $T$, perturbation radius $\rho$, hyper-parameters $\gamma_1, \gamma_2, \eta, \alpha$.
2: **Initialize:** $\theta_u \leftarrow \theta$
3: **for** $i = 1$ to $T$ **do**
4:      Sample $B_f^{(i)} \subset D_f, B_r^{(i)} \subset D_r$
5:      Calculate $\ell = \ell_f(\theta_u; B_f^{(i)}) + \ell_r(\theta_u; B_r^{(i)})$
6:      $g = \nabla_\theta \ell(\theta_u; B_f^{(i)}, B_r^{(i)})$
7:      $\delta \leftarrow \rho \cdot \frac{\nabla_\theta \ell(\theta_u; B_f^{(i)}, B_r^{(i)})}{\|\nabla_\theta \ell(\theta_u; B_f^{(i)}, B_r^{(i)})\|_2}$
8:      Calculate $\ell_f' = \ell_f(\theta_u + \delta; B_f^{(i)}), \ell_r' = \ell_r(\theta_u + \delta; B_r^{(i)})$
9:      $g_f = \nabla_\theta \ell_f'(\theta_u + \delta; B_f^{(i)}), g_r = \nabla_\theta \ell_r'(\theta_u + \delta; B_r^{(i)})$
10:      **for all** gradients $g_f', g_r'$ in $\texttt{zip}(g_f, g_r)$ **do**            ▷ Gradient consistency control
11:          **if** $g_f'$ or $g_r'$ is $\texttt{None}$ **then**
12:              **continue**                                 ▷ Skip frozen weights
13:          **end if**
14:          Compute cosine similarity:

$$\cos(g_f', g_r') = \frac{g_f'^\top g_r'}{\|g_f'\|_2^2 \cdot \|g_r'\|_2^2 + \epsilon}$$

15:          **if** $\cos(g_f', g_r') > 0$ **then**
16:              $\ell_{sim} += \frac{1}{2}\left(\|g_r' - g\|_2^\alpha + \|g_f' - g\|_2^\alpha\right)$                 ▷ GSC
17:          **else**
18:              $\ell_{diff} += \frac{1}{2}\left(\|g_r' - g\|_2^\alpha + \|g_f' - g\|_2^\alpha\right)$                ▷ GSC
19:          **end if**
20:      **end for**
21:      $g' \leftarrow \nabla_\theta(\ell + \gamma_1 \ell_{sim} + \gamma_2 \ell_{diff})$                          ▷ HA
22:      $\theta_u \leftarrow \theta_u - \eta g'$
23: **end for**
24: **return** $\theta_u$

---

### E.2  LEMMA 1 (THIRD-ORDER EXPANSION OF SMOOTHED GRADIENT)

**Lemma 2.** *Under assumptions (A1)–(A3), for any $\ell \in \{\ell_f, \ell_r\}$, we have*

$$\mathbb{E}_\delta[\nabla\ell(\theta + \delta)] = \nabla\ell(\theta) + \frac{1}{2}\, T_\ell(\Sigma) + R_\ell,$$

*where the contraction between the third-order derivative tensor and the covariance is defined component-wise as*

$$[T_\ell(\Sigma)]_k := \sum_{i,j} \frac{\partial^3 \ell(\theta)}{\partial\theta_i\partial\theta_j\partial\theta_k}\, \Sigma_{ij},$$

*and the residual satisfies*

$$\|R_\ell\| \leq C'_\ell \|\Sigma\|^{3/2},$$

*where the constant $C'_\ell$ depends on $M_\ell$ and the constant $C_3$ in assumption (A2).*

*Proof.* We expand $\nabla\ell(\theta + \delta)$ around $\theta$ up to second order (i.e., a second-order expansion of the gradient, equivalent to a third-order expansion of $\ell$):

$$\nabla\ell(\theta + \delta) = \nabla\ell(\theta) + H_\ell(\theta)\,\delta + \frac{1}{2}\left(\nabla^3\ell(\theta)\right)[\delta, \delta] + R_\ell^{(3)}(\delta),$$

where $H_\ell(\theta) = \nabla^2\ell(\theta)$, and $\left(\nabla^3\ell(\theta)\right)[\delta, \delta]$ denotes the contraction of the third-order tensor $\nabla^3\ell(\theta)$ along the directions $\delta, \delta$, yielding a vector (with components $\sum_{i,j}\partial^3_{ijk}\ell(\theta)\,\delta_i\delta_j$). The residual $R_\ell^{(3)}(\delta)$ contains terms of order three and higher. By the general form of Taylor's remainder, we have

$$\|R_\ell^{(3)}(\delta)\| \leq \frac{1}{6}\sup_{\theta' \in \mathcal{N}}\|\nabla^4\ell(\theta')\|_{\mathrm{op}}\,\|\delta\|^3.$$

In practice, we only require a coarse bound, so we use the boundedness of third-order derivatives and the upper bound on $\mathbb{E}\|\delta\|^3$ to estimate the overall residual. Since $\mathbb{E}[\delta] = 0$, the expectation of $H_\ell(\theta)\,\delta$ vanishes. Taking expectations yields

$$\mathbb{E}_\delta[\nabla\ell(\theta + \delta)] = \nabla\ell(\theta) + \frac{1}{2}\,\mathbb{E}_\delta\left[(\nabla^3\ell(\theta))[\delta, \delta]\right] + \mathbb{E}_\delta[R_\ell^{(3)}(\delta)].$$

By the covariance identity, we have component-wise

$$\mathbb{E}_\delta\left[(\nabla^3\ell(\theta))[\delta, \delta]\right]_k = \sum_{i,j}\partial^3_{ijk}\ell(\theta)\,\mathbb{E}[\delta_i\delta_j] = \sum_{i,j}\partial^3_{ijk}\ell(\theta)\,\Sigma_{ij} = [T_\ell(\Sigma)]_k.$$

Thus we obtain the main expansion, with the residual $R_\ell := \mathbb{E}_\delta[R_\ell^{(3)}(\delta)]$ bounded as

$$\|R_\ell\| \leq \mathbb{E}\|R_\ell^{(3)}(\delta)\| \leq \frac{1}{6}\sup_{\theta'}\|\nabla^4\ell(\theta')\|_{\mathrm{op}}\,\mathbb{E}\|\delta\|^3 \leq C'_\ell\|\Sigma\|^{3/2}.$$

The constant $C'_\ell$ is determined by $\sup\|\nabla^4\ell\|$ and the constant in (A2). If only third-order boundedness is assumed, a higher-order expansion yields a similar $O(\|\Sigma\|^{3/2})$ bound. This completes the proof. $\qquad\square$

### E.3  THEOREM 3 (CHANGE OF GRADIENT INNER PRODUCT AND FLIP CONDITION)

**Theorem 3.** *Under assumptions (A1)–(A3), let*

$$g_f^s := \mathbb{E}_\delta[\nabla\ell_f(\theta + \delta)], \qquad g_r^s := \mathbb{E}_\delta[\nabla\ell_r(\theta + \delta)].$$

*Then the change in the inner product*

$$\Delta := \langle g_f^s, g_r^s \rangle - \langle g_f, g_r \rangle$$

*can be expanded as*

$$\Delta = \frac{1}{2}\langle T_f(\Sigma), g_r \rangle + \frac{1}{2}\langle T_r(\Sigma), g_f \rangle + \frac{1}{4}\langle T_f(\Sigma), T_r(\Sigma) \rangle + E_R,$$

*where the residual satisfies*

$$|E_R| \leq C_1 \|T_f(\Sigma)\| \|R_r\| + C_2 \|T_r(\Sigma)\| \|R_f\| + C_3 \|R_f\| \|R_r\|,$$

*and hence there exists a constant C (depending on $M_f, M_r, C'_f, C'_r$) such that*

$$|\Delta| \leq C\big(\|g_f\| \|\Sigma\| + \|g_r\| \|\Sigma\| + \|\Sigma\|^2 + \|\Sigma\|^{3/2}\big).$$

*Moreover, if the original inner product $\langle g_f, g_r \rangle > 0$ and*

$$\frac{1}{2}\big(\|T_f(\Sigma)\|\|g_r\| + \|T_r(\Sigma)\|\|g_f\|\big) + \frac{1}{4}\|T_f(\Sigma)\|\|T_r(\Sigma)\| + o(\|\Sigma\|) \; > \; \langle g_f, g_r \rangle,$$

*then $\langle g_f^s, g_r^s \rangle < 0$, i.e., a direction flip occurs after smoothing.*

*Proof.* By Lemma 2, write

$$g_f^s = g_f + \frac{1}{2}T_f + R_f, \qquad g_r^s = g_r + \frac{1}{2}T_r + R_r,$$

with shorthand $T_f := T_f(\Sigma)$, $T_r := T_r(\Sigma)$. Substituting and expanding:

$$\begin{aligned}
\langle g_f^s, g_r^s \rangle &= \langle g_f + \tfrac{1}{2}T_f + R_f, \; g_r + \tfrac{1}{2}T_r + R_r \rangle \\
&= \langle g_f, g_r \rangle + \tfrac{1}{2}\langle T_f, g_r \rangle + \tfrac{1}{2}\langle T_r, g_f \rangle + \tfrac{1}{4}\langle T_f, T_r \rangle \\
&\quad + \langle R_f, g_r \rangle + \langle g_f, R_r \rangle + \tfrac{1}{2}\langle T_f, R_r \rangle + \tfrac{1}{2}\langle R_f, T_r \rangle + \langle R_f, R_r \rangle.
\end{aligned}$$

Thus $\Delta$ equals the above minus $\langle g_f, g_r \rangle$, where the first three terms are the main contributions and the rest are grouped as $E_R$. Using Cauchy–Schwarz to bound $E_R$ gives

$$|E_R| \leq \|R_f\| \|g_r\| + \|g_f\| \|R_r\| + \tfrac{1}{2}\|T_f\| \|R_r\| + \tfrac{1}{2}\|R_f\| \|T_r\| + \|R_f\| \|R_r\|.$$

Applying $\|R_\ell\| \leq C'_\ell \|\Sigma\|^{3/2}$ and $\|T_\ell(\Sigma)\| \leq M_\ell \|\Sigma\|$ yields the coarse overall bound

$$|\Delta| \leq C\big(\|g_f\| \|\Sigma\| + \|g_r\| \|\Sigma\| + \|\Sigma\|^2 + \|\Sigma\|^{3/2}\big).$$

Finally, if $\langle g_f, g_r \rangle > 0$ and the main correction terms dominate in the **opposite direction**, then $\langle g_f^s, g_r^s \rangle < 0$. This gives the stated sufficient condition for flipping. In other words, under graph unlearning with the two-loss paradigm, the interaction terms across losses and the cross-term components can cause the perturbed gradient directions of the two losses to reverse, i.e., $\langle g_f^s, g_r^s \rangle = \Delta + \langle g_f, g_r \rangle < 0$. When $\langle g_f, g_r \rangle \leq 0$, direction flipping is obvious. Proof complete. $\qquad \square$

### E.4 ON THE THEORETICAL SUPPRESSION BY GSC AND HA

The expansions above reveal how GSC and HA suppress the main correction terms:

- **GSC (Gradients Separation Control):** By applying distinct perturbation covariances $\Sigma_f, \Sigma_r$ or regularization weights to $\ell_f$ and $\ell_r$, the norms of $T_f(\Sigma_f), T_r(\Sigma_r)$ (and their cross term) are reduced, thereby controlling

$$\tfrac{1}{2}\langle T_f(\Sigma_f), g_r \rangle + \tfrac{1}{2}\langle T_r(\Sigma_r), g_f \rangle + \tfrac{1}{4}\langle T_f(\Sigma_f), T_r(\Sigma_r) \rangle.$$

- **HA (Human-aware Alignment):** By distinguishing "aligned/unaligned" samples and scaling the correction for inconsistent samples with a factor $s \in (0, 1)$, the dominant correction terms causing flips are attenuated. Formally, replacing $\|T_\ell(\Sigma)\|$ with $s\|T_\ell(\Sigma)\|$ in the flip condition moves the system into the safe region.

Figure 5 experimentally demonstrates that RUNNER, through its gradient consistency control mechanism, can effectively suppress the growth of these terms. Now we present the formal proof.

**Lemma 4** (Bound of third-order contraction by GSC/HA observable). *Let $\ell_f, \ell_r : \mathbb{R}^d \to \mathbb{R}$ be thrice differentiable in a neighborhood of $\theta$, and let $g_f = \nabla \ell_f(\theta)$, $g_r = \nabla \ell_r(\theta)$. For $\delta \sim (0, \Sigma)$ satisfying (A1)–(A3), define*

$$\varepsilon_f := \mathbb{E}_\delta\big[\|\nabla \ell_f(\theta + \delta) - \nabla \ell_f(\theta)\|^2\big], \quad \varepsilon_r := \mathbb{E}_\delta\big[\|\nabla \ell_r(\theta + \delta) - \nabla \ell_r(\theta)\|^2\big].$$

*Then there exists a constant $C_0 > 0$ (depending only on the boundedness of the third/fourth derivatives and $C_3$ in (A2)) such that*

$$\|T_f(\Sigma)\| \leq 2\sqrt{\varepsilon_f} + C_0 \|\Sigma\|^{3/2}, \qquad \|T_r(\Sigma)\| \leq 2\sqrt{\varepsilon_r} + C_0 \|\Sigma\|^{3/2}.$$

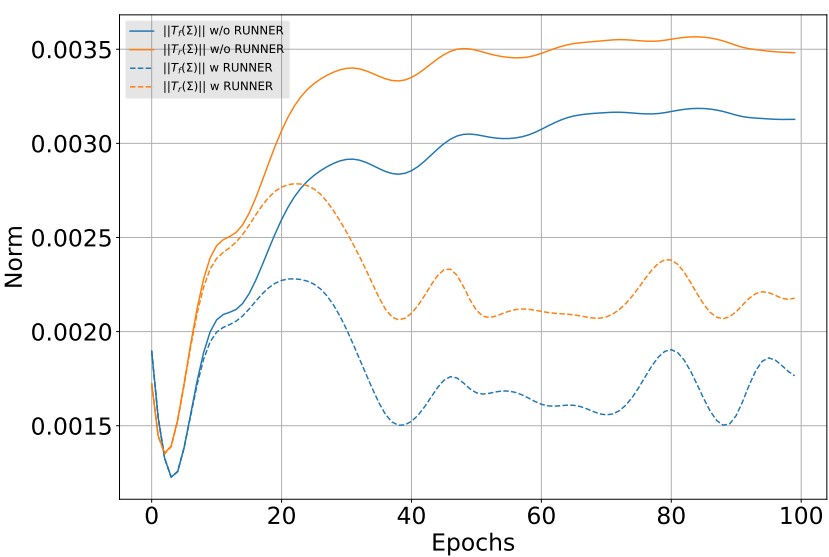

Figure 5: The impact of using RUNNER on the norms of $T_f(\Sigma_f)$ and $T_r(\Sigma_r)$.

*Proof.* From the third-order Taylor expansion (Appendix E, Lemma 2), we have

$$\mathbb{E}_\delta[\nabla\ell_f(\theta + \delta) - \nabla\ell_f(\theta)] = \tfrac{1}{2}T_f(\Sigma) + R_f,$$

where $\|R_f\| = O(\|\Sigma\|^{3/2})$. Taking norms and using Jensen's inequality,

$$\tfrac{1}{2}\|T_f(\Sigma)\| \leq \mathbb{E}\|\nabla\ell_f(\theta + \delta) - \nabla\ell_f(\theta)\| + \|R_f\| \leq \sqrt{\varepsilon_f} + \|R_f\|.$$

Hence $\|T_f(\Sigma)\| \leq 2\sqrt{\varepsilon_f} + 2\|R_f\| \leq 2\sqrt{\varepsilon_f} + C_0\|\Sigma\|^{3/2}$. The same argument holds for $T_r(\Sigma)$. □

**Theorem 5** (Lower bound of $\Delta$ and non-flip condition under GSC/HA). *Let $g_{sf}, g_{sr}$ be the smoothed gradients and define*

$$\Delta := \langle g_{sf}, g_{sr}\rangle - \langle g_f, g_r\rangle.$$

*Then*

$$\Delta \geq -\tfrac{1}{2}\big(\|T_f(\Sigma)\|\|g_r\| + \|T_r(\Sigma)\|\|g_f\|\big) - \tfrac{1}{4}\|T_f(\Sigma)\|\|T_r(\Sigma)\| - C_{\mathrm{res}}\|\Sigma\|^{3/2},$$

*for some constant $C_{\mathrm{res}} > 0$ depending on remainder terms. Combining Lemma 4, if*

$$\langle g_f, g_r\rangle > \frac{1}{2}\Big[(2\sqrt{\varepsilon_f} + C_0\|\Sigma\|^{3/2})\|g_r\| + (2\sqrt{\varepsilon_r} + C_0\|\Sigma\|^{3/2})\|g_f\|\Big]$$

$$+ \frac{1}{4}(2\sqrt{\varepsilon_f} + C_0\|\Sigma\|^{3/2})(2\sqrt{\varepsilon_r} + C_0\|\Sigma\|^{3/2}) + C_{\mathrm{res}}\|\Sigma\|^{3/2},$$

*then $\Delta > 0$, i.e. no inner-product sign flip occurs after smoothing.*

*Proof.* Expanding

$$\Delta = \tfrac{1}{2}\langle T_f, g_r\rangle + \tfrac{1}{2}\langle T_r, g_f\rangle + \tfrac{1}{4}\langle T_f, T_r\rangle + E_R,$$

with $E_R = \langle R_f, g_r\rangle + \langle R_r, g_f\rangle + \langle R_f, R_r\rangle$, we obtain by Cauchy–Schwarz

$$\Delta \geq -\tfrac{1}{2}\|T_f\|\|g_r\| - \tfrac{1}{2}\|T_r\|\|g_f\| - \tfrac{1}{4}\|T_f\|\|T_r\| - |E_R|.$$

Since $\|R_f\|, \|R_r\| = O(\|\Sigma\|^{3/2})$, we can bound $|E_R| \leq C_{\mathrm{res}}\|\Sigma\|^{3/2}$. Substituting the bounds from Lemma 4 for $\|T_f\|, \|T_r\|$ yields the stated inequality. Clearly, under the condition that $< g_r, g_f > < 0$, the goal of optimization is no longer to prevent a sign flip, but rather to control the deterioration of the inconsistency. □

The theoretical proof above shows that in the two-loss paradigm of graph unlearning, GSC/HA prevent a gradient direction flip by controlling the lower bound of $\Delta$. The experimental results in Figure 2(b) and 3(b) also confirm that this method is effective in controlling gradient inconsistency.

## F GRAPH NODE UNLEARNING

For the graph node unlearning experiments, we directly add noise to **the features of the node** to be unlearned during the inference phase to test the unlearning capability. As shown in Table 5 and 6, RUNNER achieves significantly better results than GNNDelete in the node unlearning task, and there is considerable space for improvement in robustness against noise attacks in node unlearning scenarios.

Table 5: Node unlearning robustness of RUNNER and GNNDelete on DBLP dataset under Laplacian noise attack (Level: 0.05–0.30). Delete number: 500. Task: Node Classification Prediction. We evaluate performance using accuracy (ACC) and macro-averaged F1 score (F1) on the test mask.

| Noise Level | GNNDelete | | RUNNER | |
|---|---|---|---|---|
| | ACC | F1 | ACC | F1 |
| Level=0.05 | 0.4512 | 0.3894 | 0.5752 | 0.4931 |
| Level=0.10 | 0.3496 | 0.3039 | 0.4512 | 0.3730 |
| Level=0.15 | 0.3284 | 0.2856 | 0.3832 | 0.3136 |
| Level=0.20 | 0.3108 | 0.2731 | 0.3436 | 0.2766 |
| Level=0.25 | 0.2748 | 0.2379 | 0.3096 | 0.2411 |
| Level=0.30 | 0.2616 | 0.2259 | 0.2780 | 0.2207 |

Table 6: Node unlearning robustness of RUNNER and GNNDelete on PubMed dataset under Gaussian noise attack (Level: 0.05–0.30). Delete number: 500. Task: Node Classification Prediction. We evaluate performance using accuracy (ACC) and macro-averaged F1 score (F1) on the test mask.

| Noise Level | GNNDelete | | RUNNER | |
|---|---|---|---|---|
| | ACC | F1 | ACC | F1 |
| 0.05 | 0.5356 | 0.5194 | 0.5536 | 0.5379 |
| 0.10 | 0.4380 | 0.4241 | 0.4560 | 0.4394 |
| 0.15 | 0.3972 | 0.3863 | 0.4132 | 0.3975 |
| 0.20 | 0.3776 | 0.3627 | 0.3964 | 0.3780 |
| 0.25 | 0.3808 | 0.3660 | 0.4032 | 0.3863 |
| 0.30 | 0.3704 | 0.3565 | 0.3872 | 0.3681 |

## G COMPARISON OF TIME AND MEMORY

The results in Table 7 show that RUNNER does not introduce significant additional memory consumption, and the training time increases only slightly compared to the baseline. This is due to the nature of adversarial training.

Table 7: Comparison of training time and memory consumption for RUNNER and GNNDelete for edge unlearning. TS denoets Training Time (s) for each epoch and MC denotes Memory Consumption (GB).

| Method | Cora | PubMed | CS | DBLP |
|---|---|---|---|---|
| RUNNER-TS | 0.3645 | 0.2710 | 0.4033 | 0.2901 |
| GNNDelete-TS | 0.2266 | 0.1289 | 0.2552 | 0.1201 |
| RUNNER-MC | 7.9523 | 7.2314 | 7.0174 | 0.6374 |
| GNNDelete-MC | 7.8872 | 7.0133 | 7.0132 | 0.6372 |

# H PARAMETER SENSITIVITY STUDY

## H.1 THE IMPACT OF $\alpha$

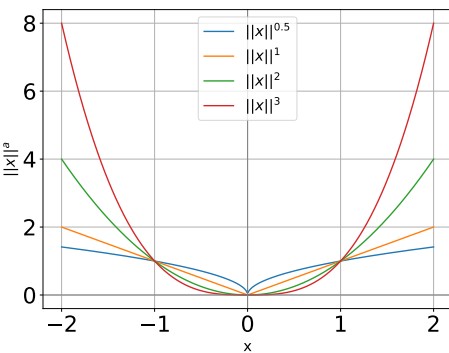

Figure 6: The Impact of $\alpha$ on the Shape of the Loss.

As shown in Figure 6, when $\alpha > 1$, it indicates a higher sensitivity to changes in $x$, which means that a unit change in $x$ will cause a larger change in $\|x\|^{\alpha}$. Therefore, using a loss function with this convex shape can suppress large fluctuations in the loss caused by weight perturbations, while allowing small fluctuations, thereby avoiding overfitting to noise. The results in Table 8 confirm the effectiveness of $\alpha > 1$ in improving unlearning robustness. The comparison of the Avg values indicates that the model's unlearning robustness is sensitive to $\alpha$, but the overall performance remains largely stable across different $\alpha$.

Table 8: Unlearning robustness comparison of RUNNER on PubMed and Cora under different $\alpha$ settings. Gaussian noise attack with level=0.05. Avg represents the average of all metrics.

| Model | PubMed | | | | Cora | | | | Avg |
|---|---|---|---|---|---|---|---|---|---|
| | $D_r$ | | $D_f$ | | $D_r$ | | $D_f$ | | |
| | AUC | AP | AUC | AP | AUC | AP | AUC | AP | |
| $\alpha = 0.5$ | 0.7856 | 0.7773 | 0.9516 | 0.9663 | 0.7907 | 0.7745 | 0.8833 | 0.9033 | 0.8291 |
| $\alpha = 1$ | 0.8083 | 0.7951 | 0.9586 | 0.9704 | 0.8201 | 0.7968 | 0.8849 | 0.8999 | 0.8418 |
| $\alpha = 2$ | 0.9392 | 0.9397 | 0.9545 | 0.9486 | 0.9138 | 0.9020 | 0.6600 | 0.5473 | 0.8256 |
| $\alpha = 3$ | 0.9496 | 0.9495 | 0.8930 | 0.8348 | 0.9217 | 0.9051 | 0.6088 | 0.5085 | 0.8089 |
| $\alpha = 5$ | 0.9504 | 0.9502 | 0.8848 | 0.8228 | 0.9224 | 0.9040 | 0.6021 | 0.5039 | 0.8051 |

## H.2 THE IMPACT OF $\gamma_1, \gamma_2$

The results in Figure 2 and Figure 3 indicate that setting $\gamma_2 > \gamma_1$ achieves better overall performance, while maintaining relatively stable comparisons across the board.

# I ADDITIONAL RESULTS

## I.1 PERFORMANCE UNDER NOISE-FREE CONDITION

The results in Table 11 demonstrate that RUNNER can effectively maintain the model's original predictive capability, i.e., its performance under no-noise attacks.

Table 9: Unlearning robustness comparison of RUNNER on PubMed and DBLP under different $\gamma$ settings. Gaussian noise attack with level=0.10. $\alpha = 2, \gamma_1 = 50$.

| Model | PubMed | | | | DBLP | | | |
| --- | --- | --- | --- | --- | --- | --- | --- | --- |
| | $D_r$ | | $D_f$ | | $D_r$ | | $D_f$ | |
| | AUC | AP | AUC | AP | AUC | AP | AUC | AP |
| $\gamma_2 = 20$ | 0.9237 | 0.9236 | 0.8277 | 0.7135 | 0.9185 | 0.9151 | 0.7284 | 0.6009 |
| $\gamma_2 = 50$ | 0.9200 | 0.9205 | 0.8565 | 0.7575 | 0.9167 | 0.9157 | 0.7331 | 0.6024 |
| $\gamma_2 = 100$ | 0.9223 | 0.9232 | 0.8769 | 0.7972 | 0.9114 | 0.9108 | 0.7412 | 0.6109 |
| $\gamma_2 = 150$ | 0.9249 | 0.9269 | 0.8888 | 0.8247 | 0.9134 | 0.9126 | 0.7540 | 0.6233 |

Table 10: Unlearning robustness comparison of RUNNER on PubMed and DBLP under different $\gamma$ settings. Gaussian noise attack with level=0.10. $\alpha = 2, \gamma_2 = 150$.

| Model | PubMed | | | | DBLP | | | |
| --- | --- | --- | --- | --- | --- | --- | --- | --- |
| | $D_r$ | | $D_f$ | | $D_r$ | | $D_f$ | |
| | AUC | AP | AUC | AP | AUC | AP | AUC | AP |
| $\gamma_1 = 50$ | 0.9249 | 0.9269 | 0.8888 | 0.8247 | 0.9134 | 0.9126 | 0.7540 | 0.6233 |
| $\gamma_1 = 150$ | 0.8975 | 0.8937 | 0.8827 | 0.8133 | 0.8955 | 0.8911 | 0.7500 | 0.6176 |
| $\gamma_1 = 200$ | 0.8871 | 0.8811 | 0.8884 | 0.8177 | 0.8946 | 0.8898 | 0.7505 | 0.6202 |

## I.2 ROBUSTNESS TO VARIOUS TYPES AND LEVELS OF NOISE

Tables 12–27 provide a detailed summary of RUNNER's performance under different datasets and various types and levels of noise attacks.

## I.3 ROBUSTNESS TO DELETION RATIOS

Tables 28–31 provide a detailed summary of RUNNER's performance under different delete ratios. The results demonstrate that it also exhibits good robustness against noise at different deletion ratios.

## I.4 ROBUSTNESS UNDER SINGLE-LOSS PERTURBATIONS

Table 32 shows that achieving robustness only in the forget loss significantly compromises robustness on the retain set, thus making the model ineffective. In the graph unlearning scenario, maintaining the stability of both losses is essential for enhancing robustness.

Table 11: Unlearning performance of RUNNER under no-noise attack. Delete Ratio: 0.5%. Task: Link Prediction. Avg represents the average of all metrics.

| Datasets | $D_r$ | | $D_f$ | | Avg |
| --- | --- | --- | --- | --- | --- |
| | AUC | AP | AUC | AP | |
| PubMed | $0.9568 \pm 0.0040$ | $0.9546 \pm 0.0040$ | $0.9632 \pm 0.0120$ | $0.9691 \pm 0.0120$ | 0.9609 |
| Cora | $0.9570 \pm 0.0020$ | $0.9534 \pm 0.0020$ | $0.9716 \pm 0.0010$ | $0.9766 \pm 0.0010$ | 0.9647 |
| DBLP | $0.9528 \pm 0.0020$ | $0.9547 \pm 0.0020$ | $0.9526 \pm 0.0040$ | $0.9557 \pm 0.0040$ | 0.9540 |
| CS | $0.9556 \pm 0.0030$ | $0.9515 \pm 0.0030$ | $0.9630 \pm 0.0020$ | $0.9707 \pm 0.0020$ | 0.9602 |

Table 12: Unlearning robustness of RUNNER on PubMed under Gaussian noise attack (Level: 0.05–0.30). Delete Ratio: 0.5%. Task: Link Prediction. Avg represents the average of all metrics.

| Model | $D_r$ | | $D_f$ | | Avg |
|---|---|---|---|---|---|
| | AUC | AP | AUC | AP | |
| Level=0.05 | $0.9414 \pm 0.0033$ | $0.9414 \pm 0.0025$ | $0.9529 \pm 0.0134$ | $0.9433 \pm 0.0242$ | 0.9448 |
| Level=0.10 | $0.9249 \pm 0.0036$ | $0.9269 \pm 0.0042$ | $0.8888 \pm 0.0205$ | $0.8247 \pm 0.0389$ | 0.8913 |
| Level=0.15 | $0.9124 \pm 0.0027$ | $0.9116 \pm 0.0045$ | $0.8081 \pm 0.0352$ | $0.6854 \pm 0.0386$ | 0.8294 |
| Level=0.20 | $0.8961 \pm 0.0055$ | $0.8918 \pm 0.0068$ | $0.6855 \pm 0.0394$ | $0.5634 \pm 0.0325$ | 0.7592 |
| Level=0.25 | $0.8772 \pm 0.0058$ | $0.8632 \pm 0.0087$ | $0.6025 \pm 0.0158$ | $0.5026 \pm 0.0095$ | 0.7114 |
| Level=0.30 | $0.8629 \pm 0.0021$ | $0.8457 \pm 0.0010$ | $0.5756 \pm 0.0310$ | $0.4866 \pm 0.0181$ | 0.6927 |

Table 13: Unlearning robustness of RUNNER on PubMed under Uniform noise attack (Level: 0.05–0.30). Delete Ratio: 0.5%. Task: Link Prediction. Avg represents the average of all metrics.

| Model | $D_r$ | | $D_f$ | | Avg |
|---|---|---|---|---|---|
| | AUC | AP | AUC | AP | |
| Level=0.05 | $0.9391 \pm 0.0065$ | $0.9345 \pm 0.0087$ | $0.5198 \pm 0.2534$ | $0.5359 \pm 0.2085$ | 0.7323 |
| Level=0.10 | $0.8969 \pm 0.0139$ | $0.8626 \pm 0.0252$ | $0.0505 \pm 0.0271$ | $0.3132 \pm 0.0048$ | 0.5308 |
| Level=0.15 | $0.8460 \pm 0.0138$ | $0.7895 \pm 0.0150$ | $0.0475 \pm 0.0115$ | $0.3122 \pm 0.0019$ | 0.4988 |
| Level=0.20 | $0.8006 \pm 0.0111$ | $0.7526 \pm 0.0068$ | $0.1122 \pm 0.0220$ | $0.3275 \pm 0.0065$ | 0.4982 |
| Level=0.25 | $0.7673 \pm 0.0078$ | $0.7283 \pm 0.0063$ | $0.1853 \pm 0.0179$ | $0.3527 \pm 0.0070$ | 0.5084 |
| Level=0.30 | $0.7421 \pm 0.0060$ | $0.7052 \pm 0.0060$ | $0.2379 \pm 0.0118$ | $0.3743 \pm 0.0051$ | 0.5149 |

Table 14: Unlearning robustness of RUNNER on PubMed under Laplacian noise attack (Level: 0.05–0.30). Delete Ratio: 0.5%. Task: Link Prediction. Avg represents the average of all metrics.

| Model | $D_r$ | | $D_f$ | | Avg |
|---|---|---|---|---|---|
| | AUC | AP | AUC | AP | |
| Level=0.05 | $0.9386 \pm 0.0031$ | $0.9393 \pm 0.0023$ | $0.9468 \pm 0.0170$ | $0.9343 \pm 0.0346$ | 0.9398 |
| Level=0.10 | $0.9198 \pm 0.0063$ | $0.9206 \pm 0.0069$ | $0.8406 \pm 0.0371$ | $0.7366 \pm 0.0594$ | 0.8544 |
| Level=0.15 | $0.9021 \pm 0.0037$ | $0.9019 \pm 0.0035$ | $0.7169 \pm 0.0276$ | $0.5905 \pm 0.0252$ | 0.7779 |
| Level=0.20 | $0.8862 \pm 0.0065$ | $0.8804 \pm 0.0087$ | $0.6561 \pm 0.0119$ | $0.5370 \pm 0.0087$ | 0.7399 |
| Level=0.25 | $0.8653 \pm 0.0076$ | $0.8561 \pm 0.0089$ | $0.6088 \pm 0.0305$ | $0.5053 \pm 0.0189$ | 0.7089 |
| Level=0.30 | $0.8456 \pm 0.0056$ | $0.8388 \pm 0.0037$ | $0.5494 \pm 0.0091$ | $0.4715 \pm 0.0047$ | 0.6763 |

Table 15: Unlearning robustness of RUNNER on PubMed under Salt-and-Pepper noise attack (Level: 0.05–0.30). Delete Ratio: 0.5%. Task: Link Prediction. Avg represents the average of all metrics.

| Model | $D_r$ | | $D_f$ | | Avg |
|---|---|---|---|---|---|
| | AUC | AP | AUC | AP | |
| Level=0.05 | $0.9468 \pm 0.0003$ | $0.9448 \pm 0.0002$ | $0.9741 \pm 0.0060$ | $0.9760 \pm 0.0094$ | 0.9604 |
| Level=0.10 | $0.9462 \pm 0.0003$ | $0.9439 \pm 0.0004$ | $0.9534 \pm 0.0091$ | $0.9442 \pm 0.0118$ | 0.9469 |
| Level=0.15 | $0.9450 \pm 0.0001$ | $0.9430 \pm 0.0001$ | $0.9355 \pm 0.0036$ | $0.9235 \pm 0.0058$ | 0.9368 |
| Level=0.20 | $0.9448 \pm 0.0005$ | $0.9431 \pm 0.0003$ | $0.9278 \pm 0.0092$ | $0.9106 \pm 0.0165$ | 0.9316 |
| Level=0.25 | $0.9442 \pm 0.0002$ | $0.9426 \pm 0.0001$ | $0.9262 \pm 0.0085$ | $0.9072 \pm 0.0126$ | 0.9301 |
| Level=0.30 | $0.9444 \pm 0.0003$ | $0.9426 \pm 0.0003$ | $0.9375 \pm 0.0045$ | $0.9222 \pm 0.0092$ | 0.9367 |

Table 16: Unlearning robustness of RUNNER on Cora under Gaussian noise attack (Level: 0.05–0.30). Delete Ratio: 0.5%. Task: Link Prediction. Avg represents the average of all metrics.

| Model | $D_r$ | | $D_f$ | | Avg |
|---|---|---|---|---|---|
| | AUC | AP | AUC | AP | |
| Level=0.05 | $0.9124 \pm 0.0105$ | $0.9005 \pm 0.0143$ | $0.6600 \pm 0.0622$ | $0.5470 \pm 0.0511$ | 0.7550 |
| Level=0.10 | $0.8722 \pm 0.0103$ | $0.8427 \pm 0.0157$ | $0.5146 \pm 0.0219$ | $0.4545 \pm 0.0110$ | 0.6710 |
| Level=0.15 | $0.8381 \pm 0.0113$ | $0.8046 \pm 0.0104$ | $0.4709 \pm 0.0090$ | $0.4352 \pm 0.0037$ | 0.6372 |
| Level=0.20 | $0.7958 \pm 0.0123$ | $0.7610 \pm 0.0117$ | $0.4627 \pm 0.0103$ | $0.4331 \pm 0.0045$ | 0.6132 |
| Level=0.25 | $0.7765 \pm 0.0073$ | $0.7408 \pm 0.0046$ | $0.4626 \pm 0.0125$ | $0.4361 \pm 0.0049$ | 0.6040 |
| Level=0.30 | $0.7540 \pm 0.0046$ | $0.7231 \pm 0.0061$ | $0.4720 \pm 0.0163$ | $0.4433 \pm 0.0080$ | 0.5981 |

Table 17: Unlearning robustness of RUNNER on Cora under Uniform noise attack (Level: 0.05–0.30). Delete Ratio: 0.5%. Task: Link Prediction. Avg represents the average of all metrics.

| Model | $D_r$ | | $D_f$ | | Avg |
|---|---|---|---|---|---|
| | AUC | AP | AUC | AP | |
| Level=0.05 | $0.8421 \pm 0.0436$ | $0.7743 \pm 0.0636$ | $0.0618 \pm 0.0328$ | $0.3150 \pm 0.0061$ | 0.4983 |
| Level=0.10 | $0.7289 \pm 0.0182$ | $0.6631 \pm 0.0127$ | $0.1856 \pm 0.0426$ | $0.3533 \pm 0.0163$ | 0.4827 |
| Level=0.15 | $0.6820 \pm 0.0095$ | $0.6275 \pm 0.0082$ | $0.2938 \pm 0.0183$ | $0.3993 \pm 0.0085$ | 0.5007 |
| Level=0.20 | $0.6572 \pm 0.0043$ | $0.6072 \pm 0.0030$ | $0.3369 \pm 0.0074$ | $0.4196 \pm 0.0036$ | 0.5052 |
| Level=0.25 | $0.6431 \pm 0.0032$ | $0.5965 \pm 0.0025$ | $0.3558 \pm 0.0035$ | $0.4287 \pm 0.0017$ | 0.5060 |
| Level=0.30 | $0.6325 \pm 0.0028$ | $0.5881 \pm 0.0022$ | $0.3652 \pm 0.0019$ | $0.4332 \pm 0.0009$ | 0.5048 |

Table 18: Unlearning robustness of RUNNER on Cora under Laplacian noise attack (Level: 0.05–0.30). Delete Ratio: 0.5%. Task: Link Prediction. Avg represents the average of all metrics.

| Model | $D_r$ | | $D_f$ | | Avg |
|---|---|---|---|---|---|
| | AUC | AP | AUC | AP | |
| Level=0.05 | $0.9122 \pm 0.0121$ | $0.8872 \pm 0.0158$ | $0.5514 \pm 0.0663$ | $0.4764 \pm 0.0395$ | 0.7068 |
| Level=0.10 | $0.8561 \pm 0.0118$ | $0.8182 \pm 0.0146$ | $0.4489 \pm 0.0140$ | $0.4262 \pm 0.0054$ | 0.6374 |
| Level=0.15 | $0.8059 \pm 0.0116$ | $0.7633 \pm 0.0123$ | $0.4494 \pm 0.0063$ | $0.4302 \pm 0.0037$ | 0.6122 |
| Level=0.20 | $0.7744 \pm 0.0043$ | $0.7321 \pm 0.0044$ | $0.4520 \pm 0.0101$ | $0.4363 \pm 0.0042$ | 0.5987 |
| Level=0.25 | $0.7655 \pm 0.0050$ | $0.7251 \pm 0.0046$ | $0.4825 \pm 0.0152$ | $0.4554 \pm 0.0062$ | 0.6071 |
| Level=0.30 | $0.7400 \pm 0.0041$ | $0.7017 \pm 0.0042$ | $0.4713 \pm 0.0100$ | $0.4558 \pm 0.0059$ | 0.5922 |

Table 19: Unlearning robustness of RUNNER on Cora under Salt-and-Pepper noise attack (Level: 0.05–0.30). Delete Ratio: 0.5%. Task: Link Prediction. Avg represents the average of all metrics.

| Model | $D_r$ | | $D_f$ | | Avg |
|---|---|---|---|---|---|
| | AUC | AP | AUC | AP | |
| Level=0.05 | $0.9390 \pm 0.0025$ | $0.9362 \pm 0.0025$ | $0.9315 \pm 0.0267$ | $0.8755 \pm 0.0604$ | 0.9206 |
| Level=0.10 | $0.9252 \pm 0.0038$ | $0.9183 \pm 0.0063$ | $0.7750 \pm 0.0466$ | $0.6456 \pm 0.0522$ | 0.8160 |
| Level=0.15 | $0.9177 \pm 0.0010$ | $0.9054 \pm 0.0016$ | $0.6717 \pm 0.0149$ | $0.5488 \pm 0.0111$ | 0.7609 |
| Level=0.20 | $0.9183 \pm 0.0017$ | $0.9057 \pm 0.0022$ | $0.6711 \pm 0.0088$ | $0.5490 \pm 0.0061$ | 0.7610 |
| Level=0.25 | $0.9199 \pm 0.0013$ | $0.9077 \pm 0.0026$ | $0.6758 \pm 0.0193$ | $0.5535 \pm 0.0134$ | 0.7642 |
| Level=0.30 | $0.9175 \pm 0.0021$ | $0.9058 \pm 0.0030$ | $0.6693 \pm 0.0188$ | $0.5470 \pm 0.0139$ | 0.7599 |

Table 20: Unlearning robustness of RUNNER on DBLP under Gaussian noise attack (Level: 0.05–0.30). Delete Ratio: 0.5%. Task: Link Prediction. Avg represents the average of all metrics.

| Model | $D_r$ | | $D_f$ | | Avg |
|---|---|---|---|---|---|
| | AUC | AP | AUC | AP | |
| Level=0.05 | $0.9453 \pm 0.0030$ | $0.9467 \pm 0.0034$ | $0.8485 \pm 0.0459$ | $0.7611 \pm 0.0765$ | 0.8754 |
| Level=0.10 | $0.9267 \pm 0.0068$ | $0.9228 \pm 0.0097$ | $0.7090 \pm 0.0427$ | $0.5818 \pm 0.0380$ | 0.7851 |
| Level=0.15 | $0.9058 \pm 0.0046$ | $0.8908 \pm 0.0080$ | $0.5579 \pm 0.0175$ | $0.4764 \pm 0.0097$ | 0.7077 |
| Level=0.20 | $0.8903 \pm 0.0022$ | $0.8654 \pm 0.0045$ | $0.5436 \pm 0.0139$ | $0.4685 \pm 0.0068$ | 0.6920 |
| Level=0.25 | $0.8718 \pm 0.0051$ | $0.8440 \pm 0.0041$ | $0.4845 \pm 0.0201$ | $0.4418 \pm 0.0086$ | 0.6605 |
| Level=0.30 | $0.8545 \pm 0.0071$ | $0.8259 \pm 0.0089$ | $0.5083 \pm 0.0119$ | $0.4549 \pm 0.0055$ | 0.6609 |

Table 21: Unlearning robustness of RUNNER on DBLP under Uniform noise attack (Level: 0.05–0.30). Delete Ratio: 0.5%. Task: Link Prediction. Avg represents the average of all metrics.

| Model | $D_r$ | | $D_f$ | | Avg |
|---|---|---|---|---|---|
| | AUC | AP | AUC | AP | |
| Level=0.05 | $0.9172 \pm 0.0226$ | $0.8989 \pm 0.0386$ | $0.3241 \pm 0.2394$ | $0.4183 \pm 0.1315$ | 0.6396 |
| Level=0.10 | $0.8172 \pm 0.0237$ | $0.7595 \pm 0.0252$ | $0.0746 \pm 0.0175$ | $0.3171 \pm 0.0043$ | 0.4921 |
| Level=0.15 | $0.7616 \pm 0.0094$ | $0.7092 \pm 0.0072$ | $0.1977 \pm 0.0418$ | $0.3582 \pm 0.0166$ | 0.5067 |
| Level=0.20 | $0.7357 \pm 0.0065$ | $0.6877 \pm 0.0071$ | $0.3104 \pm 0.0209$ | $0.4072 \pm 0.0098$ | 0.5353 |
| Level=0.25 | $0.7170 \pm 0.0045$ | $0.6681 \pm 0.0047$ | $0.3598 \pm 0.0087$ | $0.4307 \pm 0.0042$ | 0.5439 |
| Level=0.30 | $0.7004 \pm 0.0039$ | $0.6503 \pm 0.0039$ | $0.3799 \pm 0.0035$ | $0.4405 \pm 0.0017$ | 0.5428 |

Table 22: Unlearning robustness of RUNNER on DBLP under Laplacian noise attack (Level: 0.05–0.30). Delete Ratio: 0.5%. Task: Link Prediction. Avg represents the average of all metrics.

| Model | $D_r$ | | $D_f$ | | Avg |
|---|---|---|---|---|---|
| | AUC | AP | AUC | AP | |
| Level=0.05 | $0.9401 \pm 0.0050$ | $0.9418 \pm 0.0057$ | $0.8275 \pm 0.0416$ | $0.7189 \pm 0.0676$ | 0.8571 |
| Level=0.10 | $0.9099 \pm 0.0106$ | $0.9002 \pm 0.0148$ | $0.6647 \pm 0.0543$ | $0.5479 \pm 0.0420$ | 0.7557 |
| Level=0.15 | $0.8772 \pm 0.0072$ | $0.8550 \pm 0.0116$ | $0.5369 \pm 0.0143$ | $0.4653 \pm 0.0069$ | 0.6836 |
| Level=0.20 | $0.8577 \pm 0.0049$ | $0.8323 \pm 0.0032$ | $0.5078 \pm 0.0107$ | $0.4525 \pm 0.0049$ | 0.6626 |
| Level=0.25 | $0.8399 \pm 0.0052$ | $0.8116 \pm 0.0036$ | $0.5019 \pm 0.0231$ | $0.4530 \pm 0.0117$ | 0.6516 |
| Level=0.30 | $0.8193 \pm 0.0081$ | $0.7989 \pm 0.0078$ | $0.5147 \pm 0.0131$ | $0.4615 \pm 0.0063$ | 0.6486 |

Table 23: Unlearning robustness of RUNNER on DBLP under Salt-and-Pepper noise attack (Level: 0.05–0.30). Delete Ratio: 0.5%. Task: Link Prediction. Avg represents the average of all metrics.

| Model | $D_r$ | | $D_f$ | | Avg |
|---|---|---|---|---|---|
| | AUC | AP | AUC | AP | |
| Level=0.05 | $0.9469 \pm 0.0018$ | $0.9447 \pm 0.0032$ | $0.8716 \pm 0.0362$ | $0.7829 \pm 0.0773$ | 0.8865 |
| Level=0.10 | $0.9320 \pm 0.0052$ | $0.9111 \pm 0.0124$ | $0.6966 \pm 0.0461$ | $0.5712 \pm 0.0406$ | 0.7777 |
| Level=0.15 | $0.9225 \pm 0.0016$ | $0.8986 \pm 0.0039$ | $0.5706 \pm 0.0240$ | $0.4840 \pm 0.0137$ | 0.7189 |
| Level=0.20 | $0.9227 \pm 0.0019$ | $0.9059 \pm 0.0054$ | $0.5629 \pm 0.0154$ | $0.4789 \pm 0.0081$ | 0.7176 |
| Level=0.25 | $0.9204 \pm 0.0029$ | $0.8973 \pm 0.0051$ | $0.5427 \pm 0.0128$ | $0.4684 \pm 0.0066$ | 0.7072 |
| Level=0.30 | $0.9218 \pm 0.0011$ | $0.8987 \pm 0.0031$ | $0.5510 \pm 0.0138$ | $0.4733 \pm 0.0070$ | 0.7112 |

Table 24: Unlearning robustness of RUNNER on CS under Gaussian noise attack (Level: 0.05–0.30). Delete Ratio: 0.5%. Task: Link Prediction. Avg represents the average of all metrics.

| Model | $D_r$ | | $D_f$ | | Avg |
|---|---|---|---|---|---|
| | AUC | AP | AUC | AP | |
| Level=0.05 | $0.9360 \pm 0.0075$ | $0.9256 \pm 0.0105$ | $0.6813 \pm 0.0742$ | $0.5699 \pm 0.0657$ | 0.7782 |
| Level=0.10 | $0.8900 \pm 0.0120$ | $0.8553 \pm 0.0197$ | $0.4783 \pm 0.0360$ | $0.4387 \pm 0.0159$ | 0.6656 |
| Level=0.15 | $0.8382 \pm 0.0109$ | $0.7864 \pm 0.0117$ | $0.3992 \pm 0.0059$ | $0.4071 \pm 0.0026$ | 0.6077 |
| Level=0.20 | $0.8066 \pm 0.0083$ | $0.7543 \pm 0.0069$ | $0.3943 \pm 0.0048$ | $0.4099 \pm 0.0026$ | 0.5913 |
| Level=0.25 | $0.7798 \pm 0.0064$ | $0.7286 \pm 0.0058$ | $0.4136 \pm 0.0090$ | $0.4227 \pm 0.0049$ | 0.5862 |
| Level=0.30 | $0.7543 \pm 0.0063$ | $0.7039 \pm 0.0050$ | $0.4216 \pm 0.0040$ | $0.4319 \pm 0.0027$ | 0.5779 |

Table 25: Unlearning robustness of RUNNER on CS under Uniform noise attack (Level: 0.05–0.30). Delete Ratio: 0.5%. Task: Link Prediction. Avg represents the average of all metrics.

| Model | $D_r$ | | $D_f$ | | Avg |
|---|---|---|---|---|---|
| | AUC | AP | AUC | AP | |
| Level=0.05 | $0.6978 \pm 0.0515$ | $0.6353 \pm 0.0442$ | $0.2658 \pm 0.1087$ | $0.3907 \pm 0.0439$ | 0.4974 |
| Level=0.10 | $0.6244 \pm 0.0059$ | $0.5757 \pm 0.0040$ | $0.4156 \pm 0.0088$ | $0.4580 \pm 0.0044$ | 0.5184 |
| Level=0.15 | $0.6137 \pm 0.0014$ | $0.5688 \pm 0.0008$ | $0.4293 \pm 0.0016$ | $0.4648 \pm 0.0008$ | 0.5192 |
| Level=0.20 | $0.6087 \pm 0.0012$ | $0.5656 \pm 0.0008$ | $0.4334 \pm 0.0009$ | $0.4668 \pm 0.0004$ | 0.5186 |
| Level=0.25 | $0.6039 \pm 0.0010$ | $0.5623 \pm 0.0007$ | $0.4364 \pm 0.0006$ | $0.4683 \pm 0.0003$ | 0.5177 |
| Level=0.30 | $0.6010 \pm 0.0005$ | $0.5603 \pm 0.0004$ | $0.4380 \pm 0.0004$ | $0.4691 \pm 0.0002$ | 0.5171 |

Table 26: Unlearning robustness of RUNNER on CS under Laplacian noise attack (Level: 0.05–0.30). Delete Ratio: 0.5%. Task: Link Prediction. Avg represents the average of all metrics.

| Model | $D_r$ | | $D_f$ | | Avg |
|---|---|---|---|---|---|
| | AUC | AP | AUC | AP | |
| Level=0.05 | $0.9213 \pm 0.0117$ | $0.8996 \pm 0.0187$ | $0.5868 \pm 0.0757$ | $0.5006 \pm 0.0517$ | 0.7271 |
| Level=0.10 | $0.8646 \pm 0.0153$ | $0.8167 \pm 0.0203$ | $0.4283 \pm 0.0238$ | $0.4179 \pm 0.0092$ | 0.6319 |
| Level=0.15 | $0.8137 \pm 0.0125$ | $0.7558 \pm 0.0131$ | $0.3979 \pm 0.0075$ | $0.4092 \pm 0.0036$ | 0.5942 |
| Level=0.20 | $0.7785 \pm 0.0076$ | $0.7219 \pm 0.0074$ | $0.3883 \pm 0.0135$ | $0.4120 \pm 0.0069$ | 0.5752 |
| Level=0.25 | $0.7531 \pm 0.0052$ | $0.7005 \pm 0.0038$ | $0.4042 \pm 0.0059$ | $0.4262 \pm 0.0039$ | 0.5710 |
| Level=0.30 | $0.7364 \pm 0.0069$ | $0.6878 \pm 0.0050$ | $0.4296 \pm 0.0108$ | $0.4428 \pm 0.0059$ | 0.5742 |

Table 27: Unlearning robustness of RUNNER on CS under Salt-and-Pepper noise attack (Level: 0.05–0.30). Delete Ratio: 0.5%. Task: Link Prediction. Avg represents the average of all metrics.

| Model | $D_r$ | | $D_f$ | | Avg |
|---|---|---|---|---|---|
| | AUC | AP | AUC | AP | |
| Level=0.05 | $0.9442 \pm 0.0049$ | $0.9233 \pm 0.0119$ | $0.7921 \pm 0.0636$ | $0.6629 \pm 0.0747$ | 0.8306 |
| Level=0.10 | $0.9203 \pm 0.0063$ | $0.8767 \pm 0.0116$ | $0.5729 \pm 0.0479$ | $0.4854 \pm 0.0268$ | 0.7138 |
| Level=0.15 | $0.8980 \pm 0.0041$ | $0.8431 \pm 0.0047$ | $0.4483 \pm 0.0115$ | $0.4249 \pm 0.0047$ | 0.6536 |
| Level=0.20 | $0.8887 \pm 0.0023$ | $0.8270 \pm 0.0031$ | $0.3825 \pm 0.0190$ | $0.3998 \pm 0.0069$ | 0.6245 |
| Level=0.25 | $0.8912 \pm 0.0037$ | $0.8341 \pm 0.0085$ | $0.4028 \pm 0.0290$ | $0.4075 \pm 0.0109$ | 0.6339 |
| Level=0.30 | $0.8953 \pm 0.0061$ | $0.8407 \pm 0.0106$ | $0.4422 \pm 0.0206$ | $0.4225 \pm 0.0085$ | 0.6502 |

Table 28: Unlearning robustness of RUNNER on PubMed under Gaussian noise attack (Level: 0.05). Delete Ratio: 1.0%,2.5%,5.0%. Task: Link Prediction. Avg represents the average of all metrics.

| Model | $D_r$ | | $D_f$ | | Avg |
|---|---|---|---|---|---|
| | AUC | AP | AUC | AP | |
| Ratio=1.0 | $0.9368 \pm 0.0070$ | $0.9401 \pm 0.0059$ | $0.8657 \pm 0.0342$ | $0.7970 \pm 0.0649$ | 0.8849 |
| Ratio=2.5 | $0.9116 \pm 0.0138$ | $0.9150 \pm 0.0143$ | $0.8085 \pm 0.0233$ | $0.7224 \pm 0.0475$ | 0.8394 |
| Ratio=5.0 | $0.8758 \pm 0.0257$ | $0.8671 \pm 0.0300$ | $0.7641 \pm 0.0228$ | $0.6729 \pm 0.0426$ | 0.7950 |

Table 29: Unlearning robustness of RUNNER on Cora under Gaussian noise attack (Level: 0.05). Delete Ratio: 1.0%,2.5%,5.0%. Task: Link Prediction. Avg represents the average of all metrics.

| Model | $D_r$ | | $D_f$ | | Avg |
|---|---|---|---|---|---|
| | AUC | AP | AUC | AP | |
| Ratio=1.0 | $0.8831 \pm 0.0204$ | $0.8552 \pm 0.0259$ | $0.5897 \pm 0.0501$ | $0.4968 \pm 0.0317$ | 0.7062 |
| Ratio=2.5 | $0.7966 \pm 0.0351$ | $0.7437 \pm 0.0387$ | $0.5730 \pm 0.0352$ | $0.4904 \pm 0.0210$ | 0.6509 |
| Ratio=5.0 | $0.7249 \pm 0.0372$ | $0.6619 \pm 0.0344$ | $0.5731 \pm 0.0253$ | $0.4959 \pm 0.0144$ | 0.6139 |

Table 30: Unlearning robustness of RUNNER on DBLP under Gaussian noise attack (Level: 0.05). Delete Ratio: 1.0%,2.5%,5.0%. Task: Link Prediction. Avg represents the average of all metrics.

| Model | $D_r$ | | $D_f$ | | Avg |
|---|---|---|---|---|---|
| | AUC | AP | AUC | AP | |
| Ratio=1.0 | $0.9360 \pm 0.0065$ | $0.9332 \pm 0.0082$ | $0.8154 \pm 0.0455$ | $0.7139 \pm 0.0678$ | 0.8496 |
| Ratio=2.5 | $0.9051 \pm 0.0143$ | $0.8801 \pm 0.0216$ | $0.7588 \pm 0.0406$ | $0.6525 \pm 0.0544$ | 0.7991 |
| Ratio=5.0 | $0.8643 \pm 0.0273$ | $0.8132 \pm 0.0396$ | $0.7293 \pm 0.0325$ | $0.6297 \pm 0.0423$ | 0.7591 |

Table 31: Unlearning robustness of RUNNER on CS under Gaussian noise attack (Level: 0.05). Delete Ratio: 1.0%,2.5%,5.0%. Task: Link Prediction. Avg represents the average of all metrics.

| Model | $D_r$ | | $D_f$ | | Avg |
|---|---|---|---|---|---|
| | AUC | AP | AUC | AP | |
| Ratio=1.0 | $0.9002 \pm 0.0176$ | $0.8688 \pm 0.0263$ | $0.6294 \pm 0.0693$ | $0.5314 \pm 0.0559$ | 0.7324 |
| Ratio=2.5 | $0.8246 \pm 0.0350$ | $0.7753 \pm 0.0458$ | $0.5787 \pm 0.0445$ | $0.5013 \pm 0.0303$ | 0.6700 |
| Ratio=5.0 | $0.7606 \pm 0.0411$ | $0.6970 \pm 0.0456$ | $0.5742 \pm 0.0279$ | $0.5027 \pm 0.0174$ | 0.6336 |

Table 32: Unlearning robustness of model with single-loss perturbations under Gaussian noise attack (Level: 0.05). Delete Ratio: 0.5%. Task: Link Prediction. Avg represents the average of all metrics.

| Dataset | $D_r$ | | $D_f$ | | Avg |
|---|---|---|---|---|---|
| | AUC | AP | AUC | AP | |
| PubMed | $0.7733 \pm 0.0025$ | $0.7728 \pm 0.0020$ | $0.8886 \pm 0.0084$ | $0.9174 \pm 0.0099$ | 0.8380 |
| Cora | $0.7801 \pm 0.0071$ | $0.7716 \pm 0.0082$ | $0.7538 \pm 0.0285$ | $0.7909 \pm 0.0255$ | 0.7741 |
| DBLP | $0.7588 \pm 0.0024$ | $0.7403 \pm 0.0024$ | $0.8702 \pm 0.0142$ | $0.9006 \pm 0.0164$ | 0.8175 |
| CS | $0.7442 \pm 0.0067$ | $0.7253 \pm 0.0073$ | $0.7046 \pm 0.0614$ | $0.7459 \pm 0.0541$ | 0.7300 |

## J  THE COMPARISON BETWEEN RUNNER AND ADVERSARIAL TRAINING BASED ON INPUT PERTURBATIONS

Adversarial training based on input perturbations Jin et al. (2020); Rong et al. (2020) achieves robustness by adding noise to the data during the training process. In contrast, we train on clean data to improve the robustness of the inference process against noise, as shown in Figure 7. Former may have the following two potential issues: (1) Generating adversarial examples during training requires prior knowledge and incurs high computational cost. (2) Training methods based on input perturbations may result in unlearned models that are unable to defend against different types and levels of noise attacks. Our model achieves robustness through weight perturbations without adding noise to the training set, thus avoiding the impact of prior knowledge and noise types. The results of RUNNER also demonstrate that RUNNER can effectively handle various types and levels of noise attacks.

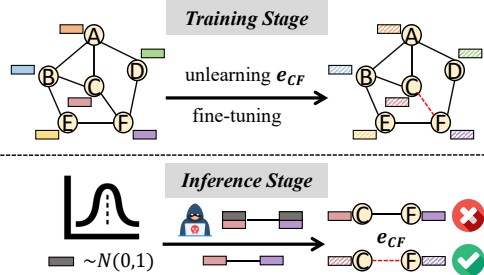

Figure 7: Robustness against noise in the inference process.

## K  THE SIGNIFICANCE AND PRACTICALITY OF THE PROBLEM

Compared to defending against Membership Inference Attacks (MIA), this work explores the trustworthiness of the unlearning process from a novel and complementary dimension—reliability. We argue that a truly trustworthy unlearning framework should not only provide privacy guarantees but also be robust and reliable.

Our robust model holds significant importance in the following practical scenarios:

**Data Correction and Dynamic Environments**: Unlearning is not solely for privacy; it is also widely used to remove low-quality, outdated, or erroneous data. In many real-world applications (e.g., financial risk control, sensor fusion graphs for autonomous driving), node input features may inherently suffer from uncertainty due to sensor errors, data acquisition noise, or environmental changes. If the unlearning process itself is highly sensitive to such natural input perturbations, its "forgetting" effect becomes unreliable and might even completely fail under certain inputs—unacceptable in safety-critical domains.

**Adversarial Probing**: Malicious attackers might subtly perturb input features to probe the boundaries and weaknesses of the unlearning process. If the effect of unlearning can be easily reversed by minor inference-time noise, this in itself constitutes a new information leakage channel, allowing attackers to infer what information the model is attempting to "forget."

Therefore, our work aims to establish robustness as a crucial pillar of "trustworthy unlearning." We believe it is as important as privacy protection, and together, they form the complete picture of unlearning technology.

## L  MORE FRAMEWORKS, DATASETS, TASKS

We conducted effectiveness evaluations of RUNNER based on the newer two-loss model MEGU, covering seven datasets and two unlearning tasks (node unlearning and edge unlearning). The unlearning ratio is 10%. Experimental results in Table 33 demonstrate that our method can still

Table 33: Experimental results of *MEGU* with *RUNNER* on different datasets.

| Dataset | Model | F1 Score ↑ – Node | F1 Score ↑ – Edge |
|---|---|---|---|
| Cora | MEGU (clean) | 84.87±1.29 | 87.54±1.01 |
| | MEGU (attack) | 72.41±0.46 | 79.88±1.10 |
| | RUNNER | **75.46±0.92** | **81.73±0.73** |
| CiteSeer | MEGU (clean) | 75.30±0.07 | 75.45±0.37 |
| | MEGU (attack) | 58.93±1.42 | 65.54±1.42 |
| | RUNNER | **59.38±1.42** | **66.96±1.65** |
| PubMed | MEGU (clean) | 86.47±0.03 | 86.37±0.16 |
| | MEGU (attack) | 68.00±2.07 | 72.85±2.37 |
| | RUNNER | **69.80±1.52** | **73.94±1.78** |
| CS | MEGU (clean) | 91.43±0.13 | 92.33±0.15 |
| | MEGU (attack) | 81.33±1.56 | 88.54±0.79 |
| | RUNNER | **81.59±1.33** | **89.01±1.21** |
| Physics | MEGU (clean) | 95.55±0.05 | 95.91±0.07 |
| | MEGU (attack) | 88.91±1.13 | 94.07±0.43 |
| | RUNNER | **90.27±1.06** | **94.39±0.37** |
| Photo | MEGU (clean) | 92.28±0.26 | 92.05±0.09 |
| | MEGU (attack) | 83.59±0.65 | 87.15±0.29 |
| | RUNNER | **85.20±1.24** | **88.16±1.17** |
| Computers | MEGU (clean) | 84.89±0.12 | 84.87±0.14 |
| | MEGU (attack) | 78.24±0.45 | 81.82±0.29 |
| | RUNNER | **79.33±0.69** | **82.55±0.74** |

achieve strong robustness under the MEGU framework, even though MEGU employs the Correct and Smooth technique to smooth prediction noise.

Table 34: Experimental results of *Cognac* with *RUNNER* on Cora dataset.

| Model | Forget Accuracy | Util Accuracy |
|---|---|---|
| oracle (clean) | 0.6925 | 0.5701 |
| poisoning attack | 0.6172 | 0.5546 |
| Cognac | 0.7106 | 0.5802 |
| Cognac + noise attack | 0.3357 | 0.2859 |
| RUNNER + noise attack | 0.4760 | 0.3601 |

We conducted Cognac with experiments against noise attacks during the inference phase. The results in Table 34 demonstrate that our RUNNER can also enhance the robustness of corrective unlearning. Additionally, based on our experimental results on Cognac, we plan to leverage subgraph similarity to reduce reliance on labels and further enhance robustness through causal reasoning.

# M MORE SCENARIOS

Table 35: Experimental results of *GNNDelete* with *RUNNER* against relearning attacks on Cora dataset.

| Model | AUC ($D_r$) | AP ($D_r$) | AUC ($D_f$) | AP ($D_f$) |
|---|---|---|---|---|
| GNNDelete | 0.9467 | 0.9467 | 0.9740 | 0.9774 |
| GNNDelete + Relearning Attack (3 epochs) | 0.9485 | 0.9470 | 0.7518 | 0.6856 |
| RUNNER + Relearning Attack (3 epochs) | 0.9320 | 0.9346 | 0.9687 | 0.9764 |

We conducted RUNNER with robustness experiments against relearning attacks (Cora dataset), based on GNNDelete. Results in Table 35 show that under relearning attacks, the performance on the forget set drops significantly after just three training epochs, whereas our RUNNER demonstrates strong robustness against such relearning attacks.

Table 36: Comparison of robustness between *MEGU* and *RUNNER* against Edge Attacks.

| Dataset | Model | F1 Score ↑ – Node | F1 Score ↑ – Edge |
|---------|-------|-------------------|-------------------|
| Cora | MEGU (clean) | 88.19±0.55 | 88.19±0.55 |
| | MEGU (attack) | 60.23±4.52 | 60.23±4.52 |
| | MEGU + RUNNER | **62.54±5.16** | **62.45±5.25** |
| CiteSeer | MEGU (clean) | 75.52±0.60 | 75.52±0.60 |
| | MEGU (attack) | 35.28±3.75 | 35.06±3.97 |
| | MEGU + RUNNER | **37.83±4.20** | **37.91±4.12** |
| Photo | MEGU (clean) | 90.88±0.29 | 90.94±0.29 |
| | MEGU (attack) | 76.92±3.33 | 76.89±3.36 |
| | MEGU + RUNNER | **77.53±4.31** | **77.50±4.34** |

In addition to feature noise attacks and relearning attacks, we have also conducted our experiments with an Edge Attack based on MEGU (randomly selecting two nodes with different labels as targets for adding noisy edges). The results across these three scenarios demonstrate that our method exhibits strong robustness under various types of attacks, although we did not specifically design it to handle adversarial perturbations.

## N    ANALYSIS OF CONFLICTING OBJECTIVE

Table 37: The result of *RUNNER* avoiding gradient flipping and mitigating conflicting optimization.

| Model | $Cos$(Pos) | $Cos$(Neg) |
|-------|-----------|-----------|
| GNNDelete | 0.84 | -0.92 |
| GNNDelete+CR | -0.64 | -0.68 |
| RUNNER ($\gamma_1 : \gamma_2 = 1:1$) | -0.62 | -0.66 |
| RUNNER ($\gamma_1 : \gamma_2 = 1:3$) | **-0.27** | **-0.54** |

We supplemented the samples for which the cosine similarity of gradients is negative due to multi-objective optimization conflict in Table 37. Although our primary focus remains on the flipping conditions for samples with positive cosine similarity induced by smoothing (as stated in Theorem 3), and on proving the effectiveness of our method, as demonstrated in Theorem 5.

Our experimental results on the Cora dataset with 0.5% edge unlearning are presented in Table 37. The number of samples with positive cosine similarity and those with negative cosine similarity is approximately 1:1. The samples with positive cosine similarity exhibit behavior fully consistent with the analysis presented in the paper on the PubMed dataset, further validating the phenomenon of smoothing-induced flipping and the effectiveness of our method. *A new finding is that*, even on the original set of samples suffering from multi-objective conflict (i.e., those with negative cosine similarity), our method can further mitigate such conflict, although the improvement is not particularly significant. Thank you for your insightful comment, which further supports and enriches our conclusion.

## O    NOISY GRAPHS

We conducted experiments on MEGU, which employs the Correct and Smooth technique for denoising on test set. The results in Table 38 show that RUNNER can further improve performance. In each dataset, the third and fourth rows present the robustness results using Correct and Smooth and

Table 38: Comparison between *MEGU* with denoising and *RUNNER* on different datasets.

| Dataset | Model | F1 Score ↑ – Node | F1 Score ↑ – Edge |
|---------|-------|-------------------|-------------------|
| Cora | MEGU (clean) | 84.87±1.29 | 87.54±1.01 |
| | MEGU (attack) | 48.80±4.33 | 54.61±4.24 |
| | MEGU (+denoising) | 72.41±0.46 | 79.88±1.10 |
| | MEGU (+RUNNER) | **75.46±0.92** | **81.73±0.73** |
| CiteSeer | MEGU (clean) | 75.30±0.07 | 75.45±0.37 |
| | MEGU (attack) | 32.50±4.27 | 34.75±5.18 |
| | MEGU (+denoising) | 58.93±1.42 | 65.54±1.42 |
| | MEGU (+RUNNER) | **59.38±1.42** | **66.96±1.65** |
| PubMed | MEGU (clean) | 86.47±0.03 | 86.37±0.16 |
| | MEGU (attack) | 59.79±2.42 | 64.02±2.73 |
| | MEGU (+denoising) | 68.00±2.07 | 72.85±2.37 |
| | MEGU (+RUNNER) | **69.80±1.52** | **73.94±1.78** |
| CS | MEGU (clean) | 91.43±0.13 | 92.33±0.15 |
| | MEGU (attack) | 66.33±5.38 | 74.97±4.89 |
| | MEGU (+denoising) | 81.33±1.56 | 88.54±0.79 |
| | MEGU (+RUNNER) | **81.59±1.33** | **89.01±1.21** |
| Photo | MEGU (clean) | 92.28±0.26 | 92.05±0.09 |
| | MEGU (attack) | 66.47±3.92 | 72.58±3.88 |
| | MEGU (+denoising) | 83.59±0.65 | 87.15±0.29 |
| | MEGU (+RUNNER) | **85.20±1.24** | **88.16±1.17** |
| Computers | MEGU (clean) | 84.89±0.12 | 84.87±0.14 |
| | MEGU (attack) | 60.61±3.07 | 65.73±3.32 |
| | MEGU (+denoising) | 78.24±0.45 | 81.82±0.29 |
| | MEGU (+RUNNER) | **79.33±0.69** | **82.55±0.74** |

RUNNER, respectively. Additionally, the comparison between the second and third rows demonstrates that denoising indeed provides a significant improvement. We also evaluated the robustness of our method in denoising capability on noisy graphs and compared it with the Cognac model. Experimental results in Table 34 demonstrate that our RUNNER performs well even in noisy graph scenarios.

## P  ADDITIONAL RESULT

Table 39: Results of hyperparameter tuning in CR.

| Model | AUC ($D_r$) | AP ($D_r$) | AUC ($D_f$) | AP ($D_f$) |
|-------|-------------|-----------|-------------|-----------|
| $\gamma_{\text{CR}} = 0.5$ | 0.9006 | 0.8930 | 0.8789 | 0.8561 |
| $\gamma_{\text{CR}} = 1$ | 0.8841 | 0.8747 | 0.8934 | 0.8860 |
| $\gamma_{\text{CR}} = 2$ | 0.8691 | 0.8584 | 0.9142 | 0.9172 |
| $\gamma_{\text{CR}} = 50$ | 0.8501 | 0.8378 | 0.9045 | 0.9059 |
| RUNNER | 0.9378 | 0.9381 | 0.9452 | 0.9320 |

The experimental results in Table 39 demonstrate that even very careful tuning of the standard smoothing hyperparameters in CR cannot achieve the performance of RUNNER.

We conducted experiments on the Cora dataset, and the results in Table 40 show that RUNNER's runtime remains on the same order of magnitude as that of most baselines.

Table 40: Experimental results of runtime for all baselines.

| Model | Time (s) |
|---|---|
| Retrain | 221 |
| GA | 21 |
| GIF | 3.87 |
| UtU | 1.71 |
| GNNDelete | 24 |
| MEGU | 13 |
| INPO | 26 |
| RUNNER | 38 |

## Q  LIMITATIONS

In Section 3.3, we made the assumption that the model is locally linear (Goodfellow et al., 2015; Szegedy et al., 2014), which is loose in practice. However, since our goal was to approximately analyze the directional effects, this assumption is acceptable. Additionally, using a smaller learning rate can make the local approximation more reasonable (Ross & Doshi-Velez, 2018). Also, our results show that the robustness of RUNNER under uniform noise attacks is not entirely satisfactory, although it represents a significant improvement compared to the baseline. This suggests that a more targeted model might be necessary to achieve better performance and robustness under such conditions.

## R  THE USE OF LARGE LANGUAGE MODELS (LLMS)

This paper utilized large language models (LLMs) for writing refinement and verification of the theorem proving process.

