# OpenReview forum: "Towards Robust Graph Unlearning via Gradient Consistency Control"
_ICLR.cc/2026/Conference — Submitted to ICLR 2026_

### Official Review · Reviewer_7aK4 · 2025-10-26

**Soundness:** 4
**Presentation:** 3
**Contribution:** 3
**Rating:** 6
**Confidence:** 3

**Summary:**

The paper targets a neglected threat in graph unlearning: tiny inference-time feature noise can undo forgetting in two-loss (forget + retain) objectives. It diagnoses the root cause as gradient inconsistency: standard robustness tricks make the forget/retain gradients point in opposing directions, derailing optimization. To fix this, the authors propose RUNNER with two parts: GSC decouples and stabilizes each loss’s gradients under weight perturbations; HA penalizes misaligned directions more than aligned ones. Across four datasets, RUNNER yields markedly better AUC/AP under multiple noise types and keeps performance under noise-free tests.

**Strengths:**

1. The paper tackles a real and neglected risk: feature noise can undo graph unlearning. It offers a clear diagnosis and a practical fix (RUNNER).

2. The empirical study is thorough and convincing: four datasets, multiple noise types and levels, clear comparisons to strong baselines, and stepwise ablations that isolate each module’s contribution.

**Weaknesses:**

1. Robustness is validated with synthetic noise injections; the study lacks evaluations on naturally noisy graph data, which may differ in structure, spectrum, and correlation patterns.

2. The paper does not compare to a denoise-first pipeline (maybe using traditional graph feature denoising methods) prior to unlearning.

**Questions:**

1. Can you test RUNNER on naturally noisy graphs to demonstrate robustness?

2. Before running graph unlearning method, what if adding a feature-denoising step? Does this help GNNDelete and RUNNER or is it unnecessary?

3. Could you add a few good-case qualitative examples where baselines fail under  noise but RUNNER succeeds, to make the benefits concrete and interpretable?

---

> ### Author Response · Authors · 2025-11-25
> **Response to Reviewer 7aK4**
>
> Thank you for your valuable and insightful comments, as well as for accurately recognizing our contributions. Moreover, we have added relevant experiments based on your suggestions. We would greatly appreciate it if you could let us know whether your concerns have been addressed in our response.
> > Q1 Can you test RUNNER on naturally noisy graphs to demonstrate robustness?
>
> ➡️We also evaluated the robustness of our method in denoising capability on noisy graphs and compared it with the *Cognac* model. Experimental results in Table K1 demonstrate that our *RUNNER* performs well even in noisy graph scenarios.
>
> **Table K1**: Experimental results of Cognac with *RUNNER* on noisy graph.
>
> |          Model          | Forget Accuracy | Util Accuracy |
> | :--: | :--: | :--: |
> |     oracle (clean)      |     0.6925      |    0.5701     |
> |    poisoning attack     |     0.6172      |    0.5546     |
> |        *Cognac*         |     0.7106      |    0.5802     |
> | *Cognac* + noise attack |     0.3357      |    0.2859     |
> | *RUNNER* + noise attack |     0.4760      |    0.3601     |
>
>
>
> > Q2 what if adding a feature-denoising step?
>
> ➡️We conducted experiments on *MEGU*, which employs the Correct and Smooth (C&S) technique for denoising on test set. The results in Table K2 show that *RUNNER* can further improve performance. In each dataset, the third and fourth rows present the robustness results using C&S and *RUNNER*, respectively. Additionally, the comparison between the second and third rows demonstrates that denoising indeed provides a significant improvement.
>
> **Table K2**: Comparison between *MEGU* with denoising and *RUNNER* on different datasets.
>
> | Dataset   | Model             | F1 Score ↑ - Node | F1 Score ↑ - Edge |
> | :-- | :-- | :--: | :--: |
> | cora      | MEGU (clean)      |    84.87±1.29     |    87.54±1.01     |
> |           | MEGU (attack)     |    48.80±4.33     |    54.61±4.24     |
> |           | MEGU (+denoising) |    72.41±0.46     |    79.88±1.10     |
> |           | MEGU (+RUNNER)    |  **75.46±0.92**   |  **81.73±0.73**   |
> | citeseer  | MEGU (clean)      |    75.30±0.07     |    75.45±0.37     |
> |           | MEGU (attack)     |    32.50±4.27     |    34.75±5.18     |
> |           | MEGU (+denoising) |    58.93±1.42     |    65.54±1.42     |
> |           | MEGU (+RUNNER)    |  **59.38±1.42**   |  **66.96±1.65**   |
> | pubmed    | MEGU (clean)      |    86.47±0.03     |    86.37±0.16     |
> |           | MEGU (attack)     |    59.79±2.42     |    64.02±2.73     |
> |           | MEGU (+denoising) |    68.00±2.07     |    72.85±2.37     |
> |           | MEGU (+RUNNER)    |  **69.80±1.52**   |  **73.94±1.78**   |
> | CS        | MEGU (clean)      |    91.43±0.13     |    92.33±0.15     |
> |           | MEGU (attack)     |    66.33±5.38     |    74.97±4.89     |
> |           | MEGU (+denoising) |    81.33±1.56     |    88.54±0.79     |
> |           | MEGU (+RUNNER)    |  **81.59±1.33**   |  **89.01±1.21**   |
> | Photo     | MEGU              |    92.28±0.26     |    92.05±0.09     |
> |           | MEGU (attack)     |    66.47±3.92     |    72.58±3.88     |
> |           | MEGU (+denoising) |    83.59±0.65     |    87.15±0.29     |
> |           | MEGU (+RUNNER)    |  **85.20±1.24**   |  **88.16±1.17**   |
> | Computers | MEGU              |    84.89±0.12     |    84.87±0.14     |
> |           | MEGU (attack)     |    60.61±3.07     |    65.73±3.32     |
> |           | MEGU (+denoising) |    78.24±0.45     |    81.82±0.29     |
> |           | MEGU (+RUNNER)    |  **79.33±0.69**   |  **82.55±0.74**   |
>
>
>
> > Q3 add a few good-case qualitative examples where baselines fail under noise but RUNNER succeeds
>
> ➡️We now provide some qualitative examples of good cases. On the **Cora** dataset, in the setting where **0.5%** of edges are unlearned, the following edges are cases where *GNNDelete* fails but *RUNNER* succeeds: $e_1 = ( 9123, 9144 )$,  $e_2 = ( 12789, 18674 )$, $e_3 = ( 16230, 17947 )$, $e_4 = ( 2298, 16281  )$, $e_5 = ( 662, 5001  )$.

---

### Official Review · Reviewer_kb2H · 2025-10-27

**Soundness:** 2
**Presentation:** 2
**Contribution:** 2
**Rating:** 4
**Confidence:** 3

**Summary:**

This work investigates the robustness of approximate graph unlearning methods (those using a unified forget-and-retain loss) against inference-time noise on node features. The authors claim that standard robustness techniques like smoothing introduce a "gradient inconsistency" between the forget and retain objectives, affecting the optimization. They propose RUNNER, a framework to stabilize gradients independently and a "human-aware alignment" (HA) loss to penalize inconsistencies during optimization. Experiments suggest RUNNER improves robustness against noise attacks compared to baseline unlearning and naive smoothing, while preserving utility on clean data.

**Strengths:**

1. **Problem:** Addresses the relevant and under-explored problem of robustness in graph unlearning against inference-time noise.
2. **New insights:** Provides empirical evidence (Figure 2) suggesting standard smoothing techniques can interact negatively with the two-loss unlearning objective.
3. **New framework:** Proposes a concrete framework combining decoupled regularization and an alignment-inspired loss (HA) to mitigate the identified issue. Demonstrates empirical improvements in robustness against various noise types compared to baseline unlearning and naive smoothing application (Table 1, Figure 3a).

**Weaknesses:**

1. **Problem Framing:** The "gradient inconsistency" might be overstated as a unique phenomenon caused by smoothing, potentially being just a byproduct of the conflicts in the two-loss objective that smoothing doesn't resolve well. Stronger evidence is needed to differentiate it from standard multi-objective optimization challenges.
2. **Theoretical Support:** The theoretical analysis relies on strong assumptions (local linearity) and provides intuition rather than rigorous guarantees about RUNNER's effectiveness or the necessity of its components.
3. **Complexity of Solution:** The HA loss component adds complexity with extra hyperparameters ($\gamma_1, \gamma_2, \alpha$) requiring careful tuning (Appendix H), potentially making the framework less practical.
4. **Narrow Scope:** Focuses primarily on inference-time noise for edge unlearning (link prediction) with GNNDelete. Robustness against other perturbations (ex., noisy unlearning requests, training noise) or for other tasks/models is not thoroughly explored.
5. **Limited Baselines:** Comparison against adversarial training methods adapted for the two-loss setting is missing. I would like to see Cognac [1] also included.

---

*[1] Kolipaka, Varshita, Akshit, Sinha, Debangan, Mishra, Sumit, Kumar, Arvindh, Arun, Shashwat, Goel, Ponnurangam, Kumaraguru. "A Cognac Shot To Forget Bad Memories: Corrective Unlearning for Graph Neural Networks." Proceedings of the 42nd International Conference on Machine Learning (ICML).*

**Questions:**

1. Can you provide more (apart from PubMed) experiments or analysis to more clearly distinguish the "gradient inconsistency exacerbated by smoothing" from the baseline level of gradient conflict inherent in optimizing the forget and retain losses simultaneously *without* smoothing? Does smoothing consistently make the cosine similarity *more* negative than without smoothing, or does it just fail to improve it?
2. The HA loss formulation seems complex. Have you experimented with simpler gradient alignment techniques, such as directly adding a penalty term like $-\cos(g_f, g_r)$ or projecting gradients to remove conflicting components (as in some multi-task learning literature)? How does HA compare?
4. How does RUNNER perform when applied to other approximate unlearning methods besides GNNDelete and INPO (MEGU, GIF, Cognac)? Does the gradient inconsistency issue manifest similarly across different two-loss formulations?
5. Could the observed robustness improvements be achieved simply by very careful tuning of the standard smoothing hyperparameters (like $\gamma$ in CR) without needing RUNNER's specific components, perhaps accepting a slight trade-off in clean performance?

---

> ### Author Response · Authors · 2025-11-25
> **Response to Reviewer kb2H (1/2)**
>
> Thank you for your valuable and insightful comments, as well as for accurately recognizing our contributions. Moreover, we have added relevant experiments based on your suggestions. We would greatly appreciate it if you could let us know whether your concerns have been addressed in our response.
> > Q1 provide more (apart from PubMed) experiments or analysis; Does smoothing consistently make the cosine similarity *more* negative than without smoothing, or does it just fail to improve it?
>
> ➡️We have now supplemented the samples for which the cosine similarity of gradients is negative due to multi-objective optimization conflict in Table H1 . Although our primary focus remains on **the flipping conditions** for samples with positive cosine similarity induced by smoothing (as stated in Theorem 3), and on proving the effectiveness of our method, as demonstrated in Theorem 5.
>
> ➡️Our experimental results on the Cora dataset with 0.5% edge unlearning are presented in the table below. The number of samples with positive cosine similarity and those with negative cosine similarity is approximately 1:1. The samples with positive cosine similarity exhibit behavior fully consistent with the analysis presented in the paper on the PubMed dataset, further validating the phenomenon of **smoothing-induced flipping** and the effectiveness of our method. *A new finding is that*, even on the original set of samples suffering from multi-objective conflict (i.e., those with negative cosine similarity), our method can further **mitigate such conflict**, although the improvement is not particularly significant. Thank you for your insightful comment, which further supports and enriches our conclusion.
>
> **Table H1**: The result of RUNNER avoiding gradient flipping and mitigating conflicting optimization.
>
> | Model                             | *Cos*(Pos) | *Cos*(Neg) |
> | -- | :-: | :-: |
> | GNNDelete                         | 0.84       | -0.92      |
> | GNNDelete+CR                      | -0.64      | -0.68      |
> | RUNNER ($\gamma_1:\gamma_2=1:1$)  | -0.62      | -0.66      |
> | RUNNER  ($\gamma_1:\gamma_2=1:3$) | **-0.27**  | **-0.54**  |
>
>
>
> > Q2 The HA loss formulation seems complex. Have you experimented with simpler gradient alignment techniques, such as directly adding a penalty term like $-cos(g_f, g_r)$ or projecting gradients to remove conflicting components (as in some multi-task learning literature)? How does HA compare?
>
> ➡️The results in Tables H2 and H3 show that, compared to HA ($\gamma_1:\gamma_2=1:3$), directly imposing gradient constraint $-cos(g_f, g_r)$ not only degrades performance in the noise-free mode but also fails to defend against noise attacks. Our primary goal is to investigate how to **avoid gradient flipping caused by smoothing techniques**, thereby achieving **robustness** in unlearning technique.
>
> **Table H2**: Results on the Cora dataset with only the regularization term $- \cos(g_f, g_r)$ added.
>
> | Model (Cora)                                  | AUC ($D_r$) | AP ($D_r$) | AUC ($D_f$) | AP ($D_f$) |    AVG     |
> | -- | :-: | :-: | :-: | :-: | :--: |
> | GNNDelete                                     | 0.8986      | 0.8745     | 0.5141      | 0.4568     |   0.6860   |
> | GNNDelete + ($-cos(g_f, g_r)$)                | 0.6508      | 0.7076     | 0.6284      | 0.7864     |   0.6933   |
> | GNNDelete + noise attack + ($-cos(g_f, g_r)$) | 0.6498      | 0.6947     | 0.5915      | 0.7310     |   0.6668   |
> | RUNNER  ($\gamma_1:\gamma_2=1:3$)             | 0.9021      | 0.8836     | 0.6132      | 0.5145     | **0.7284** |
>
>
>
> **Table H3**: Results on the PubMed dataset with only the regularization term $- \cos(g_f, g_r)$ added.
>
> | Model (PubMed)                                | AUC ($D_r$) | AP ($D_r$) | AUC ($D_f$) | AP ($D_f$) |    AVG     |
> | -- | :-: | :-: | :-: | :-: | :--: |
> | GNNDelete                                     | 0.9464      | 0.9460     | 0.7584      | 0.6244     |   0.8188   |
> | GNNDelete + ($-cos(g_f, g_r)$)                | 0.6394      | 0.6993     | 0.5893      | 0.7597     |   0.6719   |
> | GNNDelete + noise attack + ($-cos(g_f, g_r)$) | 0.6397      | 0.6984     | 0.5786      | 0.7480     |   0.6662   |
> | RUNNER  ($\gamma_1:\gamma_2=1:3$)             | 0.9378      | 0.9381     | 0.9452      | 0.9320     | **0.9383** |

---

> ### Author Response · Authors · 2025-11-25
> **Response to Reviewer kb2H (2/2)**
>
> >  Q3 How does RUNNER perform when applied to other approximate unlearning methods besides GNNDelete and INPO (MEGU, GIF, Cognac)? Does the gradient inconsistency issue manifest similarly across different two-loss formulations?
>
> ➡️We have added experiments based on MEGU and Cognac, and the results in Table H4 and H5 demonstrate that our RUNNER remains effective under both models, which adopt different two-loss formulations. GIF modifies the weights in a single step based on influence functions, which differs from our post-training approach.
>
> **Table H4**: Experimental results of MEGU with RUNNER on different datasets.
>
> | Dataset (ratio=0.1) | Model         | F1 Score ↑ - Node | F1 Score ↑ - Edge |
> | :-- | :-- | :--: | :--: |
> | cora                | MEGU (clean)  |    84.87±1.29     |    87.54±1.01     |
> |                     | MEGU (attack) |    72.41±0.46     |    79.88±1.10     |
> |                     | RUNNER        |  **75.46±0.92**   |  **81.73±0.73**   |
> | citeseer            | MEGU          |    75.30±0.07     |    75.45±0.37     |
> |                     | MEGU (attack) |    58.93±1.42     |    65.54±1.42     |
> |                     | RUNNER        |  **59.38±1.42**   |  **66.96±1.65**   |
> | pubmed              | MEGU          |    86.47±0.03     |    86.37±0.16     |
> |                     | MEGU (attack) |    68.00±2.07     |    72.85±2.37     |
> |                     | RUNNER        |  **69.80±1.52**   |  **73.94±1.78**   |
> | CS                  | MEGU          |    91.43±0.13     |    92.33±0.15     |
> |                     | MEGU (attack) |    81.33±1.56     |    88.54±0.79     |
> |                     | RUNNER        |  **81.59±1.33**   |  **89.01±1.21**   |
> | Physics             | MEGU          |    95.55±0.05     |    95.91±0.07     |
> |                     | MEGU (attack) |    88.91±1.13     |    94.07±0.43     |
> |                     | RUNNER        |  **90.27+1.06**   |  **94.39±0.37**   |
> | Photo               | MEGU          |    92.28±0.26     |    92.05±0.09     |
> |                     | MEGU (attack) |    83.59±0.65     |    87.15±0.29     |
> |                     | RUNNER        |  **85.20±1.24**   |  **88.16±1.17**   |
> | Computers           | MEGU          |    84.89±0.12     |    84.87±0.14     |
> |                     | MEGU (attack) |    78.24±0.45     |    81.82±0.29     |
> |                     | RUNNER        |  **79.33±0.69**   |  **82.55±0.74**   |
>
> **Table H5**: Experimental results of Cognac with RUNNER on Cora dataset.
>
> |         Model         | Forget Accuracy | Util Accuracy |
> | :--: | :--: | :--: |
> |    oracle (clean)     |     0.6925      |    0.5701     |
> |   poisoning attack    |     0.6172      |    0.5546     |
> |        Cognac         |     0.7106      |    0.5802     |
> | Cognac + noise attack |     0.3357      |    0.2859     |
> | RUNNER + noise attack |     0.4760      |    0.3601     |
>
>
>
> >  Q4 Could the observed robustness improvements be achieved simply by very careful tuning of the standard smoothing hyperparameters (like in CR) without needing RUNNER's specific components, perhaps accepting a slight trade-off in clean performance?
>
> ➡️The experimental results in Table H6 demonstrate that even very careful tuning of the standard smoothing hyperparameters in CR cannot achieve the performance of RUNNER.
>
> **Table H6**: Results of hyperparameter tuning in CR.
>
> | Model               | AUC ($D_r$) | AP ($D_r$) | AUC ($D_f$) | AP ($D_f$) |
> | :-: | :-: | :-: | :-: | :-: |
> | $\gamma_{CR} = 0.5$ | 0.9006      | 0.8930     | 0.8789      | 0.8561     |
> | $\gamma_{CR} = 1$   | 0.8841      | 0.8747     | 0.8934      | 0.8860     |
> | $\gamma_{CR} = 2$   | 0.8691      | 0.8584     | 0.9142      | 0.9172     |
> | $\gamma_{CR} = 50$  | 0.8501      | 0.8378     | 0.9045      | 0.9059     |
> | RUNNER              | 0.9378      | 0.9381     | 0.9452      | 0.9320     |

---

### Official Review · Reviewer_exEx · 2025-10-30

**Soundness:** 2
**Presentation:** 3
**Contribution:** 2
**Rating:** 4
**Confidence:** 3

**Summary:**

The paper diagnoses a lack of robustness of recent two-loss graph unlearning methods (e.g., GNNDelete / INPO) to inference-time feature noise and proposes RUNNER, which (i) separately regularizes forget/retain gradients and (ii) adds a gradient-alignment penalty. Empirically, RUNNER shows strong improvements on link-prediction (edge-unlearning) tasks on four datasets. The paper contains theoretical claims about why naive smoothing increases gradient inconsistency.
The topic is important and the core idea (control gradient interactions between competing unlearning objectives) is sensible and has empirical merit. However, the work has several weaknesses to address. Therefore, I am currently on the borderline and look forward to a strong rebuttal by the authors, following which I am happy to revise my score.

**Strengths:**

- The problem setup defined is interesting, and is a refreshing change from the privacy preservation objective of unlearning.
- The motivation is clear. The gradient-inconsistency phenomenon is demonstrated empirically (cosine similarity plots) and used as the basis for the method.
- The experiments study multiple noise families and ablate components.
- The proposed method is simple and elegantly solves the proposed problem.

**Weaknesses:**

1. I would like to see more of a discussion on the paradigm of corrective unlearning introduced in [1]. The motivation of noise injection is similar to adversarial attacks, and could warrant a deeper discussion and also an inclusion as a baseline, as [1] has similar experiments.
2. The experiments are severely lacking in many areas.
    - Breadth and validity of datasets. There are only 4 datasets discussed, and all of them are old datasets which the community has been warning against [2]. I feel in general the experimental breadth is not enough to support the paper's claims. PubMed should absolutely not be used to present the main findings of the paper. Similarly, node unlearning results are deferred to the appendix and are only compare with GNNDelete on two datasets. I understand the motivation of the paper is focused on two-loss methods, however comparison with a wide variety of methods is necessary to evaluate if this problem is actually a problem in practice.
    - Questionable performance for retrain. Retraining is usually the gold standard for unlearning. Why is the performance low in Table 1? Is this due to the added noise? Also, retraining performance is not present everywhere. Please be consistent with the methods reported.
    - Lack of error bars. The paper reports the mean performance, but does not report the standard deviation in the runs, making it hard to identify the statistical significance of the work.
   - Comparison with newer methods. Why have newer methods like [1, 3] not been evaluated? In general the evaluation seems limited. It would be nice to see if newer methods also suffer from robustness issues and would make the paper's claims stronger.
   - Runtime analysis. Please provide a runtime analysis for all methods in your main results, not just GNNDelete, which is at this point a very old baseline.
3. Is robustness only a problem for two-loss methods? It would be nice to see if other methods suffer from it or not, particularly [1] since it operates on adversarial attacks.
4. It would help the paper's merit if the authors can discuss attacker capabilities and whether the defense generalizes to stronger adversarial attacks (worst-case adversarial perturbations crafted with full model knowledge) as opposed to random noise. If the method only helps against random noise but not adversarial perturbations, that should be clearly stated.

[1] https://arxiv.org/abs/2412.00789

[2] https://arxiv.org/abs/2502.14546

[3] https://arxiv.org/abs/2401.11760

**Questions:**

Please see weaknesses.

---

> ### Author Response · Authors · 2025-11-25
> **Response to Reviewer exEx (1/2)**
>
> Thank you for your valuable and insightful comments, as well as for accurately recognizing our contributions. Moreover, we have added relevant experiments based on your suggestions. We would greatly appreciate it if you could let us know whether your concerns have been addressed in our response.
> > Q1 more of a discussion on the paradigm of corrective unlearning introduced in Cognac;
> >
> > Q2.4  Comparison with newer methods Cognac.
>
> ➡️Difference: *Cognac* aims to use corrective unlearning techniques to remove the adverse effects of manipulated training data (such as Feature Poisoning Attack) on GNN predictions. This attack occurs during the **training phase**. Our *RUNNER* primarily trains a **robust** unlearning model to defend against attacks that occur during the **inference** or **post-training** phases, such as evasion attacks (e.g., adding noise) and latest relearning attacks. The application scenarios of *Cognac* also demonstrate the value of our robustness enhancement in eliminating noisy or outdated data.
>
> ➡️Similarity: Both *Cognac* and *RUNNER* adopt a competitive two-loss paradigm.
>
>
>
> ➡️We have supplemented *Cognac* with experiments against noise attacks during the inference phase. The results in Table E1 demonstrate that our *RUNNER* can also enhance the robustness of corrective unlearning.
>
> **Table E1**: Experimental results of *Cognac* with *RUNNER* on Cora dataset.
>
> |         Model         | Forget Accuracy | Util Accuracy |
> | :--: | :--: | :--: |
> |    oracle (clean)     |     0.6925      |    0.5701     |
> |   poisoning attack    |     0.6172      |    0.5546     |
> |        Cognac         |     0.7106      |    0.5802     |
> | Cognac + noise attack |     0.3357      |    0.2859     |
> | RUNNER + noise attack |     0.4760      |    0.3601     |
>
>
>
> ➡️We have also supplemented *RUNNER* with robustness experiments against **relearning attacks** (Cora dataset), based on *GNNDelete*. Results in Table E2 show that under relearning attacks, the performance on the forget set drops significantly after just three training epochs, whereas our *RUNNER* demonstrates strong robustness against such relearning attacks.
>
> **Table E2**: Experimental results of *GNNDelete* with *RUNNER* against relearning attacks on Cora dataset.
>
> | Model                                    | AUC ($D_r$) | AP ($D_r$) | AUC ($D_f$) | AP ($D_f$) |
> | -- | :--: | :--: | :--: | :-: |
> | GNNDelete                                |   0.9467    |   0.9467   |   0.9740    |   0.9774   |
> | GNNDelete + Relearning Attack (3 epochs) |   0.9485    |   0.9470   |   0.7518    |   0.6856   |
> | RUNNER + Relearning Attack (3 epochs)    |   0.9320    |   0.9346   |   0.9687    |   0.9764   |
>
>
>
> > Q2.1 Breadth and validity of datasets;
> >
> > Q2.3 Lack of error bars;
> >
> > Q2.4 Comparison with newer methods MEGU.
>
> ➡️We conducted effectiveness evaluations of *RUNNER* based on the newer two-loss model *MEGU*, covering **seven** datasets and **two** unlearning tasks (node unlearning and edge unlearning). The unlearning ratio is 10%. Experimental results in Table E3 demonstrate that our method can still achieve strong robustness under the *MEGU* framework, even though *MEGU* employs the **Correct and Smooth** technique to smooth prediction noise.
>
> **Table E3**: Experimental results of *MEGU* with *RUNNER* on different datasets.
>
> | Dataset   | Model         | F1 Score ↑ - Node | F1 Score ↑ - Edge |
> | :-- | :-- | :--: | :--: |
> | cora      | MEGU (clean)  |    84.87±1.29     |    87.54±1.01     |
> |           | MEGU (attack) |    72.41±0.46     |    79.88±1.10     |
> |           | RUNNER        |  **75.46±0.92**   |  **81.73±0.73**   |
> | citeseer  | MEGU (clean)  |    75.30±0.07     |    75.45±0.37     |
> |           | MEGU (attack) |    58.93±1.42     |    65.54±1.42     |
> |           | RUNNER        |  **59.38±1.42**   |  **66.96±1.65**   |
> | pubmed    | MEGU (clean)  |    86.47±0.03     |    86.37±0.16     |
> |           | MEGU (attack) |    68.00±2.07     |    72.85±2.37     |
> |           | RUNNER        |  **69.80±1.52**   |  **73.94±1.78**   |
> | CS        | MEGU (clean)  |    91.43±0.13     |    92.33±0.15     |
> |           | MEGU (attack) |    81.33±1.56     |    88.54±0.79     |
> |           | RUNNER        |  **81.59±1.33**   |  **89.01±1.21**   |
> | Physics   | MEGU (clean)  |    95.55±0.05     |    95.91±0.07     |
> |           | MEGU (attack) |    88.91±1.13     |    94.07±0.43     |
> |           | RUNNER        |  **90.27+1.06**   |  **94.39±0.37**   |
> | Photo     | MEGU (clean)  |    92.28±0.26     |    92.05±0.09     |
> |           | MEGU (attack) |    83.59±0.65     |    87.15±0.29     |
> |           | RUNNER        |  **85.20±1.24**   |  **88.16±1.17**   |
> | Computers | MEGU (clean)  |    84.89±0.12     |    84.87±0.14     |
> |           | MEGU (attack) |    78.24±0.45     |    81.82±0.29     |
> |           | RUNNER        |  **79.33±0.69**   |  **82.55±0.74**   |

---

> > ### Author Response · Authors · 2025-11-25
> > **Response to Reviewer exEx (2/2)**
> >
> > > Q2.2 Questionable performance for retrain.
> >
> > ➡️In graph unlearning, the reason why retraining using only the retained set leads to low prediction performance on the forget set is due to the presence of graph topology, which causes predictions on these untrained forget nodes/edges to be close to random guessing. This observation is consistent with the metrics reported by *GNNDelete* and *INPO*. Additionally, *MEGU* does not directly evaluate performance on the forget set, so its reported results are very high—serving as a gold standard.
> >
> >
> >
> > > Q2.5 Runtime analysis.
> >
> > ➡️We conducted experiments on the Cora dataset, and the results in Table E4 show that *RUNNER*'s runtime remains on the same order of magnitude as that of most baselines.
> >
> > **Table E4**: Experimental results of runtime for all baselines.
> >
> > | Model     | time (s) |
> > | -- | -- |
> > | Retrain   | 221      |
> > | GA        | 21       |
> > | GIF       | 3.87     |
> > | UtU       | 1.71     |
> > | GNNDelete | 24       |
> > | MEGU      | 13       |
> > | INPO      | 26       |
> > | RUNNER    | 38       |
> >
> >
> >
> > > Q3 Is robustness only a problem for two-loss methods?
> >
> > ➡️Our work mainly focuses on the fact that when the cross-term between the higher-order correction terms $T_f$ and $T_r$ of the two competing losses becomes too large, it can cause a **gradient flip** (As stated in Theorem 3). Theorem 5 shows that our *RUNNER* can control the lower bound to prevent gradient flipping. Our results in Table E1 demonstrate that our robustness is also effective in scenarios defending against adversarial attacks on training stage.
> >
> >
> >
> > > Q4 discuss attacker capabilities.
> >
> > ➡️In addition to feature noise attacks, we have also supplemented our experiments with an Edge Attack based on *MEGU* (randomly selecting two nodes with different labels as targets for **adding noisy edges**). Moreover, the results in Table E2 also demonstrate that *RUNNER* performs very well in defending against **relearning attacks**.
> >
> > **Table E5:** Comparison of robustness between *MEGU* and *RUNNER* against Edge Attacks.
> >
> > | Dataset  | Model         | F1 Score ↑ - Node | F1 Score ↑ - Edge |
> > | :-- | :-- | :--: | :--: |
> > | cora     | MEGU (clean)  |    88.19±0.55     |    88.19±0.55     |
> > |          | MEGU (attack) |    60.23±4.52     |    60.23±4.52     |
> > |          | MEGU + RUNNER |  **62.54±5.16**   |  **62.45±5.25**   |
> > | citeseer | MEGU (clean)  |    75.52±0.60     |    75.52±0.60     |
> > |          | MEGU (attack) |    35.28±3.75     |    35.06±3.97     |
> > |          | MEGU + RUNNER |  **37.83±4.20**   |  **37.91±4.12**   |
> > | Photo    | MEGU (clean)  |    90.88±0.29     |    90.94±0.29     |
> > |          | MEGU (attack) |    76.92±3.33     |    76.89±3.36     |
> > |          | MEGU + RUNNER |  **77.53±4.31**   |  **77.50±4.34**   |

---

> > > ### Comment · Reviewer_exEx · 2025-11-25
> > >
> > > Thank you for the detailed rebuttal. My concerns have been addressed. I have consequently revised my score.

---

> > > > ### Author Response · Authors · 2025-11-26
> > > > **Response to Reviewer exEx**
> > > >
> > > > We sincerely appreciate your thoughtful consideration of our rebuttals. Your valuable suggestions and insightful comments have enabled our method to be validated across a broader range of datasets, models and application scenarios, significantly reinforcing the necessity and importance of our framework. We are truly grateful for your engagement and for recognizing the contributions of our work!

---

### Author Response · Authors · 2025-11-28
**Summary of Rebuttal Revision**

We sincerely thank all the reviewers for their time and efforts to review our work. In response to the valuable and insightful feedback, we have made several updates to our manuscript, as outlined below:

1. We have added the discussion on the paradigm of corrective unlearning introduced in *Cognac* and included *Cognac* and *MEGU* as baselines for comparison. Additionally, inspired by the novel scenario introduced in Cognac, we have also included further discussions based on the results of *Cognac* + *RUNNER* .
2. We have added new experimental results based on the *MEGU* framework, including **seven** additional datasets and **two** unlearning tasks, to further demonstrate the breadth and effectiveness of our method.
3. We have added **three** new scenario-based experimental results, including Feature Poisoning Attacks, Edge Attacks, and Relearning Attacks, to demonstrate the broader applicability of our method. We have also provided a clearer statement regarding the capability of our method to defend against attacks.
4. We have further elaborated on the theoretical support. For simplicity in the main text, we adopted the "locally linear" assumption to facilitate explanation; however, the proof of Theorem 3 in Appendix E is based on a more rigorous third-order Taylor expansion, and the conclusion regarding **gradient flip does not rely on the linear assumption**.
5. We have added experiments to demonstrate that gradient flip is not caused by conflicting optimization objectives, thereby further empirically validating Theorem 3 on another dataset.
6. We have added experiments on naturally **noisy graphs** and the **feature-denoising** step to further demonstrate the effectiveness of our method.
7. We have also included additional experimental results regarding model training time, parameter tuning, directly applying the $- \cos(g_f, g_r)$ term, and qualitative examples in favorable cases.

We hope that the revised manuscript can help address the concerns and resolve the issues raised by the reviewers.  The revised portions have been highlighted in blue text, including main text and Appendix.



Best,

Authors of Submission 5144

---

### Author Response · Authors · 2025-11-30
**Rebuttal Summary For AC**

We sincerely thank AC for your time and effort to review our work. We have summarized our rebuttal to the reviewers as follows:

1. We have added the discussion on the paradigm of corrective unlearning introduced in *Cognac*. Moreover, we introduced **two new two-loss frameworks** (*Cognac* and *MEGU* ) as baselines to validate the effectiveness of our method. (For reviewers $\textcolor{blue}{exEx}$ and $\textcolor{blue}{kb2H}$)
2. We have added new experimental results based on the *MEGU* framework, including **seven additional datasets** and **two unlearning tasks**, to further demonstrate the breadth and effectiveness of our method. (For all reviewers)
3. We have added **three new scenario**-based experimental results, including Feature Poisoning Attacks, Edge Attacks, and Relearning Attacks, to demonstrate the broader applicability of our method. We have also provided a clearer statement regarding the capability of our method to defend against attacks.  (For reviewers $\textcolor{blue}{exEx}$ and $\textcolor{blue}{kb2H}$)
4. We further clarified that the linear assumption is not essential to our theory. For simplicity in the main text, we adopted the "locally linear" assumption to facilitate explanation; however, the proof of Theorem 3 in Appendix E is based on a more rigorous third-order Taylor expansion, and the conclusion regarding **gradient flip does not rely on the linear assumption**.  (For reviewer $\textcolor{blue}{kb2H}$)
5. We have added experiments to demonstrate that gradient flip is not caused by conflicting optimization objectives on another dataset, thereby further empirically validating Theorem 3. (For reviewer $\textcolor{blue}{kb2H}$)
6. We have added experiments on naturally **noisy graphs** and the **feature-denoising** step to further demonstrate the effectiveness of our method. (For reviewer $\textcolor{blue}{7aK4}$)
7. We have also included additional experimental results regarding model training time, parameter tuning, directly applying the $- \cos(g_f, g_r)$ term, and qualitative examples in favorable cases. (For all reviewers)



We have also provided a revised version for you and reviewers. We hope that the revised manuscript can help address the concerns and resolve the issues raised by the reviewers.  The revised part have been highlighted in $\textcolor{blue}{blue}$ text, including main text and Appendix.



Finally, we believe our work can inspire the community to pay more attention to the robustness of unlearning algorithms when removing harmful, outdated, or noisy data, ensuring that forgetting cannot be easily reversed or ineffective.



Best,

Authors of Submission 5144

---

### Meta-Review · Area_Chair_uMZV · 2026-01-07

**Summary:**

All reviewers (exEx, kb2H, 7aK4) expressed concerns regarding the limited scope of experiments, emphasizing the use of only older, smaller datasets (Pubmed, Cora), which reduces confidence in generalization and practical relevance.

The initial submission lacked sufficient baseline comparisons, particularly omitting newer and stronger methods. Reviewers exEx and kb2H explicitly highlighted the lack of comprehensive baseline comparisons, particularly with newer methods such as Cognac, MEGU, and GIF, weakening the evaluation's credibility.

Reviewer kb2H pointed out that the theoretical analysis relied heavily on strong assumptions (local linearity) and lacked rigorous guarantees, thus weakening confidence in the method's theoretical soundness.

The robustness experiments mainly use synthetic noise injections, raising concerns regarding the realism and applicability of findings in real-world noisy scenarios.
Reviewers exEx and kb2H noted the robustness evaluations were superficial, primarily involving simple synthetic noise injections, without adequate exploration of advanced adversarial attacks or realistic noisy scenarios.

Reviewer kb2H emphasized concerns about the practical complexity introduced by additional hyperparameters and alignment mechanisms, questioning the real-world applicability and ease-of-use of the proposed approach.

**Reviewer Concerns:**

The authors’ rebuttal partially addressed several experimental concerns by adding evaluations on newer baselines (Cognac, MEGU) and expanding to additional datasets (Reviewers exEx, kb2H). However, the most significant issues remain unresolved:
* Insufficient baseline comparisons: While some newer methods were included, the overall baseline coverage and initial positioning remain insufficient (Reviewers exEx, kb2H).
* Theoretical weaknesses: The theoretical justification continues to rely on strong assumptions and lacks rigorous support, significantly limiting the claimed theoretical contributions (Reviewers kb2H, exEx).
* Synthetic robustness experiments: Although the authors provided some evaluation on naturally noisy data, the majority of robustness validations still heavily depend on artificial perturbations, questioning their real-world relevance (Reviewer 7aK4, kb2H).
* Complexity and practical applicability: The added experiments did not convincingly address concerns regarding the practicality and complexity of the proposed framework (Reviewers kb2H, exEx).

**Reviewer Scores:**

Despite addressing certain minor experimental points, the authors failed to resolve the reviewers' core concerns about limited baselines, outdated datasets, overly simplified theoretical assumptions, and insufficient comprehensive analyses. Thus, the reviewers are likely to maintain their original scores, and the overall recommendation remains reject.

---

### Decision · Program_Chairs · 2026-01-26

Reject